# Influence and Predictors of Anxiety on Health Status ≥ 5 Years Beyond Breast Cancer Diagnosis in Spain: A Cross-Sectional Study

**DOI:** 10.3390/life15060932

**Published:** 2025-06-10

**Authors:** Francisco Álvarez-Salvago, Clara Pujol-Fuentes, Jose Medina-Luque, Maria Figueroa-Mayordomo, Carmen Boquete-Pumar, Sandra Atienzar-Aroca

**Affiliations:** 1FIBIO Research Group, Department of Physiotherapy, Faculty of Health Sciences, European University of Valencia, 46010 Valencia, Spain; francisco.alvarez2@universidadeuropea.es (F.Á.-S.); jose.med.luque@gmail.com (J.M.-L.); maria.figueroa@universidadeuropea.es (M.F.-M.); sandra.atienza@universidadeuropea.es (S.A.-A.); 2Department of Health Sciences, Faculty of Health Sciences, University of Jaén, 23071 Jaén, Spain; cbp00029@red.ujaen.es; 3Department of Languages, Arts and Education, Faculty of Education, Antonio de Nebrija University, 28248 Madrid, Spain; 4Department of Dentistry, Faculty of Health Sciences, European University of Valencia, 46010 Valencia, Spain

**Keywords:** anxiety, cognitive functioning, breast cancer, long-term survivorship, cancer-related fatigue, rehabilitation

## Abstract

**Purpose:** To explore the association between anxiety levels and health outcomes in long-term breast cancer survivors (LTBCSs) and to identify predictors of anxiety in this population. **Methods**: A multicenter cross-sectional study was conducted with 80 LTBCSs, categorized into two groups based on their anxiety levels: low anxiety (≤3.4) and high anxiety (≥3.5). The analysis focused on variables assessed at least five years after diagnosis, including sociodemographic and clinical data, mood, cancer-related fatigue (CRF), pain, self-perceived physical fitness, physical activity (PA), and health-related quality of life (HRQoL). ANOVA, Mann–Whitney U, and chi-square tests were conducted, along with correlation and multiple regression analysis. Effect sizes were calculated using Cohen’s *d*. **Results**: Among the participants, 46.25% exhibited higher anxiety levels. This group showed significantly worse mood, self-perceived physical fitness, and HRQoL and elevated CRF and pain (*p* < 0.05). Regression analysis identified “total CRF” (*β* = 0.51; *p* < 0.01) and “cognitive functioning” (*β* = −0.24; *p* = 0.02) as significant predictors of higher levels of anxiety (r^2^ adjusted = 0.470). **Conclusions**: Anxiety significantly impacts multiple dimensions of health in LTBCSs. Total CRF and cognitive functioning are key predictors of anxiety. These findings have direct clinical implications: routine psychological and physical assessments should be integrated into survivorship care to identify individuals at risk and inform targeted interventions to enhance long-term well-being and HRQoL.

## 1. Introduction

Breast cancer (BC) is the most commonly diagnosed malignancy among women worldwide [1], and advances in early detection and treatment have significantly increased survival rates [2]. As a result, attention has increasingly shifted toward understanding the long-term sequelae experienced by BC survivors, particularly those who have surpassed the five-year mark following diagnosis and primary treatment. These long-term breast cancer survivors (LTBCSs) often face a complex array of persistent physical, emotional, and psychological symptoms that may compromise their health-related quality of life (HRQoL) and overall well-being [3,4].

Among these psychological concerns faced by LTBCSs, anxiety remains one of the most prevalent yet under-characterized conditions [5,6,7]. Despite its clinical significance, it has often been grouped with depression under the broader umbrella of psychological distress [4,8,9]. However, recent evidence suggests that anxiety may involve distinct physiological and behavioral mechanisms [6,7,10,11,12,13] and may also follow a unique trajectory, particularly in response to psychological treatment, exhibiting differential responses to specific therapeutic approaches such as cognitive–behavioral therapy or mindfulness-based interventions [14]. This distinction is further supported by evidence of potential biopsychosocial pathways through which anxiety may influence health outcomes in BC survivors, including neuroendocrine dysregulation, maladaptive health behaviors, and heightened inflammatory responses, which together justify the focused assessment of anxiety in this population [15].

Studies specifically focusing on anxiety in both BC survivors [16,17] and LTBCSs [5,18] have shown that high anxiety levels are associated with poorer coping mechanisms, diminished cognitive functioning, greater cancer-related fatigue (CRF), and reduced HRQoL. Nevertheless, many existing studies do not provide an integrative analysis of factors associated with anxiety in LTBCSs, which limits the design of targeted interventions. Methodological heterogeneity and inconsistent outcome measures further hinder the interpretation and generalization of findings [5,16,17,18]. A clearer understanding of anxiety and its predictors in this population is therefore essential to develop personalized, multidisciplinary care strategies. Recognizing anxiety as a distinct clinical concern may help improve long-term outcomes, as it remains frequently underdiagnosed despite its substantial impact on HRQoL and functional recovery.

Considering the aforementioned, the present study aims to (1) explore the association between anxiety levels and health outcomes in LTBCSs and (2) identify predictors of anxiety in this population ≥ 5 years beyond BC diagnosis. By focusing solely on anxiety, this study seeks to fill a critical gap in survivorship research and support the development of more individualized interventions aimed to improve the long-term emotional and physical well-being of LTBCSs.

## 2. Materials and Methods

### 2.1. Study Design and Sample

This study followed a cross-sectional and descriptive design and was conducted in accordance with the Declaration of Helsinki (version 14/2017) [19]. Ethical approval was obtained from the Biomedical Research Ethics Committee of Granada (CEIm) (Ref: 1038-N-16 I.P/26 July 2018). A total of 80 LTBCSs were recruited in 2024 through the oncology department of the San Cecilio University Hospital of Granada and various BC associations in Granada and Málaga. A detailed description of the participant recruitment process is provided in Appendix A.

Sample size was calculated using G*Power (Version 3.1.9.7), assuming a medium effect size (f = 0.25; Cohen’s *d* = 0.50), α = 0.05, and power = 0.80. This assumption is supported by studies on anxiety and related outcomes [20,21] and by similar sample size calculation approaches used in cancer survivor populations for other clinical variables [22]. A minimum of 72 participants was required, and 80 were ultimately recruited. Evaluations were conducted either at the Faculty of Health Sciences (University of Granada) or within the facilities of the collaborating associations.

The oncology department was aware of the study’s inclusion and exclusion criteria for women referred from the hospital. When a potential participant was identified, the oncologist informed her about the study and provided our contact number. If she was interested, she voluntarily reached out to us, ensuring data protection. Upon contact, further details were provided via phone, an in-person appointment was scheduled at the university, and a study dossier was offered. Any questions were addressed, and if the participant agreed, she signed the informed consent form before undergoing an on-site assessment lasting approximately 50 min.

For participants from BC associations, a research team member visited the associations to present the study. Interested women were identified by the association and contacted us voluntarily to schedule an assessment. The same protocol used for hospital-referred participants was applied for assessments at the association premises.

The same researcher conducted all assessments, regardless of location. Additionally, one team member was responsible for digitalizing the questionnaires, while another, independently, analyzed the results.

Eligibility criteria included being at least 18 years old, having been diagnosed with stage I–IIIa BC at least five years prior, having no current psychiatric disorders, not undergoing active cancer treatment (radiotherapy, chemotherapy, and/or hormonal therapy), and being Spanish-speaking. Participants using psychotropic medication or those unable to understand or complete the assessments were excluded from the study.

Anxiety levels were measured to classify participants into groups using an adapted version of the Visual Analogue Scale (VAS) for Anxiety (VAS-A). This scale consists of a 10 cm horizontal line ranging from “no anxiety” (0) on the left to “worst possible anxiety” (10) on the right. Although the concept of a VAS adapted to measure anxiety was proposed in 1976 and first applied in 1988 for patients undergoing dental treatment [23], it has since demonstrated utility in various clinical populations, including women with BC [24]. More recent studies have shown a significant correlation between VAS-A scores and those obtained with standardized instruments like the Hospital Anxiety and Depression Scale (HADS) or the State-Trait Anxiety Inventory (STAI), suggesting its validity as a rapid screening tool for anxiety [25,26]. As such, it may serve as an efficient preliminary screening tool for anxiety and potential anxiety symptoms, particularly in clinical contexts where time constraints preclude the use of more comprehensive measures. Therefore, and following similar recommendations for cut-off points [25], participants were classified into two groups: those with low anxiety levels (≤3.4) and those with higher anxiety levels (≥3.5).

### 2.2. Measures

#### 2.2.1. Primary Outcome

The level of anxiety was evaluated using an adapted version of the VAS. The VAS-A is a 10 cm scale where 0 signifies “no anxiety” and 10 represents the “worst possible anxiety” [23]. Participants were asked to rate their level of anxiety at the time of the assessment. Previous studies have shown this tool to be reliable for assessing anxiety in BC patients [24].

#### 2.2.2. Secondary Outcomes

##### Demographic and Clinical Data Collection

Structured interviews were conducted using a tailored questionnaire to gather sociodemographic and clinical information. Clinical variables assessed included time since diagnosis, tumor stage, family history of BC, surgical interventions, types of treatment undergone, current medication use, presence of metastasis or recurrence, menopausal status, engagement with psychological or physiotherapy services, and lifestyle habits such as smoking and alcohol consumption.

##### Mood State

Mood state was assessed using the Scale for Mood Assessment (EVEA), which has demonstrated high reliability, with a Cronbach’s alpha ranging from 0.88 to 0.93 [27]. This instrument consists of 16 items, each rated on a Likert scale from 0 to 10, and evaluates four dimensions: sadness/depression, anxiety, anger/hostility, and happiness. The score for each dimension is obtained by averaging the corresponding item scores, with higher values indicating greater intensity of the respective mood state.

##### Cancer-Related Fatigue

The Piper Fatigue Scale (PFS), a 22-item instrument, was used to assess CRF across four key dimensions: behavioral severity, affective, sensory, and cognitive/mood. The total score reflects overall CRF intensity, with higher values indicating greater fatigue [28,29]. The reliability of the scale has been confirmed (Cronbach’s alpha = 0.86) [30]. Previous research supports two classification models (Models A and B) for CRF severity [28,29]. In Model A, CRF is categorized as follows: 0 = none, 1–3 = mild, 4–6 = moderate, and 7–10 = severe. Model B defines the categories as: 0 = none, 1–2 = mild, 3–5 = moderate, and 6–10 = severe. Notably, moderate CRF—regardless of the model used—is considered clinically relevant [31]. Patients with moderate to severe CRF should undergo further evaluation to identify any underlying or comorbid conditions requiring medical attention [32].

##### Pain

Pain intensity was evaluated using the original version of the VAS, a 10 cm scale known for its high reliability with an intraclass correlation coefficient (ICC = 0.97) [33], where 0 signifies “no pain” and 10 represents the “worst possible pain”. Participants were asked to rate the pain in both their affected and unaffected arms at the time of the assessment. For those with bilateral BC where both arms were impacted, the arm deemed “affected” was identified based on these factors: (1) the patient’s subjective pain report when comparing the two arms, (2) the level of surgical intervention, and (3) the presence of lymphedema or other post-surgical complications. Additionally, the Brief Pain Inventory (BPI) short form, with Cronbach’s alpha values ranging from 0.87 to 0.89 [34], was employed to measure pain intensity (severity) through four questions, and its impact on daily life (interference) through seven questions. Higher scores reflected greater pain intensity and more significant interference with daily activities.

##### Self-Perceived Physical Fitness

Self-perceived physical fitness was assessed using the International Fitness Scale (IFIS), a questionnaire that employs a 5-point Likert scale ranging from 1 (very poor) to 5 (very good). This instrument includes five key items evaluating general fitness, as well as specific components such as cardiorespiratory endurance, muscular strength, speed/agility, and flexibility in comparison to peers. The IFIS has shown moderate reliability, with an average weighted kappa 0.45 [35].

##### Physical Activity Level

PA was assessed using the Minnesota Leisure Time Physical Activity (MLTPA) questionnaire, which evaluates the average frequency and total hours dedicated to PA over the past week. This tool has shown high reliability, with an ICC of 0.95 [36]. The assessment was based on a predefined list of specific physical activities. To calculate energy expenditure, the reported weekly duration (in hours) of each activity was multiplied by its corresponding metabolic equivalent of task (MET) value [37], which reflects the energy cost of the activity. Higher scores indicate a greater total weekly PA duration. For the main analysis, the total PA value obtained was categorized into three groups based on previously established cutoff points: very low activity (≤3 MET), low activity (3.1–7.4 MET), and sufficiently active (≥7.5 MET) [38,39,40].

##### Health-Related Quality of Life

HRQoL was evaluated using two validated instruments: the EORTC QLQ-C30 (version 3.0) and its BC-specific module, the QLQ-BR23. These tools have demonstrated reliability, with Cronbach’s alpha values ranging from 0.46 to 0.94 [41]. Responses were recorded on a 4-point Likert scale (1 = not at all, 4 = very much) and subsequently converted to a 0–100 scale. In the interpretation of results, higher scores on functional and global HRQoL scales reflect better health status, whereas higher scores on symptom scales indicate a greater burden of symptoms. Additionally, a summary score for the QLQ-C30 was derived by aggregating 13 scales and items, excluding global health status and financial impact scales. For this summary score, greater values represent better overall HRQoL [42].

### 2.3. Statistical Analysis

Data analysis was performed using the Statistical Package for the Social Sciences (IBM SPSS Statistics for Windows, version 27.0, Armonk, NY, USA). The significance threshold was set at *p* < 0.05, with a 95% confidence interval (CI). The Kolmogorov–Smirnov test was applied to assess the normality of all variables, considering *p* > 0.05 as indicative of normal distribution.

Normally distributed variables (age, time since diagnosis, and time since first surgery) were analyzed using ANOVA to compare the two anxiety-level groups: low (≤3.4) vs. high (≥3.5). Results are reported as mean ± standard deviation. Post hoc comparisons were adjusted using Bonferroni correction to account for multiple testing.Non-normally distributed variables (mood state, non-categorized CRF, pain, self-perceived physical fitness, and HRQoL) were analyzed using the Mann–Whitney U test. Results are also reported as mean ± standard deviation. Bonferroni correction was applied where appropriate to minimize the risk of Type I error due to multiple comparisons.Categorical variables, including other demographic, clinical, and medical characteristics, as well as categorized CRF and PA levels (based on predefined cut-off scores) [28,29,38,39,40], were analyzed using chi-square tests and are presented as percentages.

Additionally, effect sizes (Cohen’s *d*) were classified as follows: negligible (*d* = 0–0.19), small (*d* = 0.2–0.49), moderate (*d* = 0.5–0.79), large (*d* = 0.8–1.19), and very large (*d* ≥ 1.20) [43].

As most variables did not meet the normality assumption, Spearman correlation analysis was performed to examine the relationship between anxiety levels (assessed with the VAS-A and analyzed as a continuous, non-categorized variable) and the other study variables. Furthermore, a stepwise multiple regression analysis was conducted to identify factors contributing to long-term anxiety variability. Variables were included in the regression model if they met two conditions: (1) a significant correlation with the dependent variable and (2) a correlation coefficient below 0.70 between independent variables to mitigate collinearity issues [44]. A forward selection approach was employed, incorporating significant predictors sequentially based on their association strength with the dependent variable. At each step, the significance of the linear regression model was evaluated, and standardized β coefficients were calculated for the final model.

## 3. Results

### 3.1. Demographic and Clinical Characteristics

No significant differences were observed between the groups in terms of the demographic and clinical characteristics of the 80 LTBCSs, based on their anxiety levels. Following established recommendations for cut-off points [25], participants were classified into two groups using the VAS-A: those with low anxiety levels (53.75%) and those with high anxiety levels (46.25%).

The mean age of participants with low anxiety levels was 48.41 ± 8.05 years, whereas for those with high anxiety levels it was 50.51 ± 8.02 years. Among individuals with low anxiety levels, 44.2% had a university-level education, 25.6% were on sick leave, 20.9% were smokers, 95.3% had received both radiotherapy and chemotherapy, and 60.5% had attended psychological therapy sessions in the past three months. In contrast, within the group of individuals with high anxiety, 24.3% had a university-level education, 51.4% were on sick leave, 27% were smokers, 81.1% had undergone both radiotherapy and chemotherapy, and 67.6% had attended psychological therapy sessions in the past three months. Additional details on demographic and clinical characteristics are provided in Table 1.

### 3.2. Mood State

The analysis of mood states, assessed using the EVEA, revealed significant differences between groups. LTBCSs with high anxiety levels reported greater “sadness–depression” and “anger–hostility”, as well as lower “happiness” compared to those with low anxiety levels (U = 310.00 to 765.00; all *p* < 0.01; *d* = 0.06 to >1.20). A detailed overview of these findings can be found in Table 2.

### 3.3. Cancer-Related Fatigue

The assessment of PFS domains revealed significant differences between the groups. LTBCSs with high anxiety levels showed higher levels of CRF across all domains compared to those with low anxiety levels (U = 323.50 to 380.00; all *p* < 0.01; *d* = 1.11 to > 1.20) (Table 2).

In relation to the two cut-off scores, A and B, the analysis indicated that LTBCSs with high anxiety levels had a significantly greater prevalence of “moderate” to “severe” CRF for both cut-off scores (A = 67.5% and B = 78.3%) than those with low anxiety levels (A = 18.6% and B = 21%) (both *p* < 0.01). For a visual representation of these differences, refer to Figure 1.

### 3.4. Pain

The evaluation of pain identified significant differences between the groups in both the VAS and BPI measures. Specifically, LTBCSs with high anxiety levels reported greater pain intensity in both the affected and non-affected arms (VAS), as well as increased pain severity and interference (BPI) compared to those with low anxiety levels (U = 434.00 to 549.50; *p* < 0.01 to 0.01; *d* = 0.71 to 0.99) (Table 2).

### 3.5. Self-Perceived Physical Fitness

The assessment of self-perceived physical fitness, measured using the IFIS, identified significant differences between the groups across all domains. LTBCSs with high anxiety levels reported lower self-perceived “general physical fitness”, “cardiorespiratory endurance”, “muscular strength”, “speed–agility”, and “flexibility” compared to those with low anxiety levels (U = 493.00 to 742.50; *p* < 0.01 to 0.05; *d* = 0.13 to 0.73) (Table 2).

### 3.6. Physical Activity Level

The analysis of between-group differences in MLTPA scores did not reveal significant differences (*p* > 0.05). However, LTBCSs with high anxiety levels exhibited a higher rate of inactivity (35.1%) compared to those with low anxiety levels (18.6%). Furthermore, only 29.7% of LTBCSs with high anxiety levels met the minimum PA recommendations [38,39,40]. Although this proportion was slightly higher among LTBCSs with low anxiety levels, it remained limited to 37.2% (refer to Table 2). Hence, higher anxiety levels may be associated with lower engagement in physical activity among LTBCSs.

### 3.7. Health-Related Quality of Life

The analysis of HRQoL, assessed using the QLQ-C30, revealed significant differences between groups. LTBCSs with high anxiety levels had significantly lower scores in “role functioning”, “emotional functioning”, “cognitive functioning”, “social functioning”, “global health status”, and the “summary score”. Additionally, they reported significantly higher scores in all symptom scales and single-item measures, except for “constipation”, compared to LTBCSs with low anxiety levels (U = 316.00 to 634.00; *p* < 0.01 to 0.04; *d* = 0.20 to 1.11). No significant differences were found in “physical functioning” between the groups (*p* > 0.05). See Table 3 for further details.

Moreover, significant differences between the groups were observed in the BR23 module. LTBCSs with high anxiety levels had significantly lower scores in “body image” and “sexual functioning”, as well as higher scores in “future perspective” and all symptom scales compared to the other group (U = 388.00 to 680.00; *p* < 0.01 to 0.01; *d* = 0.18 to 0.90). No significant differences were found in “sexual enjoyment” between the groups (*p* > 0.05) (refer to Table 3).

Therefore, considering both the QLQ-C30 and BR23, higher anxiety levels in LTBCSs appears to be linked to poorer HRQoL across functional and symptom domains.

### 3.8. Correlation Analysis and Multiple Regression Analysis

The Spearman’s correlation analysis revealed significant positive correlations between the level of anxiety and the following variables: PFS: “behavioral/severity”, “affective”, “sensory”, “cognitive”, and “total fatigue score”, QLQ-C30: “fatigue”, “nausea and vomiting”, “pain”, “dyspnea”, “insomnia”, “appetite loss”, and “financial difficulties”, QLQ-BR23: “breast symptoms”, “arm symptoms”, and “upset by hair loss”, VAS: “affected arm” and “non-affected arm”, BPI: “intensity” and “interference” (ρ = 0.231 to 0.587; *p* < 0.01 to 0.03). Meanwhile, significant negative correlations were observed between the level of anxiety and the following variables: “role functioning”, “emotional functioning”, “cognitive functioning”, “social functioning”, “global health”, and “summary score”, QLQ-BR23: “body image”, “sexual functioning”, and “systemic therapy side effects”, IFIS: “general physical fitness”, “speed/agility”, and “flexibility” (ρ = −0.302 to −0.550; all *p* < 0.01). In summary, higher anxiety levels are generally associated with greater CRF, symptom burden, and pain, alongside reduced functioning, global HRQoL, and self-perceived physical fitness. The results are presented in Figure 2.

The final regression model determined that “total CRF” from the PFS (*β* = 0.051; *p* = <0.01) and “cognitive functioning” from the QLQ-C30 (*β* = −0.024; *p* = 0.02) were significant predictors of high anxiety levels. Collectively, these variables accounted for 47.0% of the variance in anxiety (*r^2^* adjusted = 0.470; *p* < 0.01 to 0.02) among individuals who were ≥5 years post-cancer diagnosis. See Table 4 for further details.

## 4. Discussion

The first aim of this study was to explore the association between anxiety levels and health outcomes in LTBCSs. The second aim was to identify factors that influence anxiety in this population. The main findings of this study revealed that 46.25% of LTBCSs exhibited high anxiety levels, while 53.75% had low anxiety levels. Those with high anxiety demonstrated greater impairments in mood, self-perceived physical fitness, and HRQoL, alongside elevated levels of CRF and pain. Additionally, the combination of “total CRF” and “cognitive functioning” accounted for 47.0% of the variance in anxiety among LTBCSs. Although the cut-off point used to define high and low anxiety (3.5 vs. 3.4) may appear narrow, it was derived from a previous validation study using the VAS-A in oncological settings [25], where scores ≥ 3.5 have been shown to reflect clinically relevant anxiety. In our cohort, this threshold revealed consistent and meaningful differences in multiple health domains. Therefore, despite the subtle numerical distinction, the stratification proved useful for identifying LTBCSs at greater risk for symptom burden and poorer health outcomes. However, this cut-off should not be interpreted as a definitive diagnostic threshold. Rather, it may serve as an initial red flag in time-constrained clinical settings, prompting more comprehensive psychological assessments or referral to mental health professionals when anxiety scores exceed 3.5.

### 4.1. Mood State

In relation to mood state, our findings indicate that LTBCSs with elevated anxiety levels experience significantly higher levels of sadness–depression and anger–hostility, alongside reduced happiness, compared to those with low anxiety levels. This observation is consistent with existing literature, which highlights that LTBCSs often face persistent psychological challenges, including anxiety and depression, even years post-treatment [5,6,7,45]. In this sense, the interplay between anxiety and mood disturbances in LTBCSs can be attributed to several factors. Elevated anxiety may stem from concerns about cancer recurrence, leading to heightened emotional distress [46]. Additionally, the physical and social challenges encountered during survivorship, such as ongoing treatment side effects and changes in social roles, can exacerbate negative mood states [47]. Addressing these psychological concerns is crucial, as unmanaged anxiety can adversely affect overall well-being and hinder the long-term recovery process.

### 4.2. Cancer-Related Fatigue

As for CRF, it is a common concern in LTBCSs and is closely linked to psychological distress, particularly anxiety and depression [3]. Consistent with previous studies, our findings show that participants with high anxiety levels reported significantly greater CRF across all PFS domains, supporting a bidirectional relationship between fatigue and mood. Shared neurobiological mechanisms—such as HPA axis dysregulation and inflammation—may underlie this association [48]. Notably, the prevalence of moderate to severe CRF was substantially higher among LTBCSs with elevated anxiety, with 67.5% and 78.3% meeting severity thresholds under models A and B, respectively. These results reinforce evidence that anxiety exacerbates fatigue and its functional impact during survivorship [49]. As moderate CRF warrants clinical attention [31], screening for anxiety may be key to improving fatigue management. Future research should investigate targeted interventions, such as cognitive–behavioral therapy or structured exercise, to address anxiety-related CRF and enhance long-term quality of life.

### 4.3. Pain

Pain is a prevalent symptom among LTBCSs, with studies reporting post-treatment pain in approximately 46% of cases [50,51]. Our results show that those with high anxiety levels report significantly greater pain intensity in both the affected and unaffected arms (VAS), as well as higher pain severity and interference scores (BPI), compared to those with low anxiety. These findings align with evidence linking anxiety to increased pain perception in post-mastectomy women [3]. This relationship may be driven by shared neurobiological pathways, such as anxiety-related amplification of pain through dysregulated dopaminergic activity and overlapping pain–emotion neural circuits [52]. Moreover, the bidirectional interplay between pain and anxiety can create a self-reinforcing cycle of emotional and physical distress. Addressing anxiety may enhance pain management and improve HRQoL in LTBCSs. Integrated interventions targeting both symptoms should be prioritized in survivorship care.

### 4.4. Self-Perceived Physical Fitness and Physical Activity Levels

With respect to self-perceived physical fitness and PA levels, they are both important components of long-term health outcomes in BC survivors. Our findings indicate that LTBCSs with high anxiety levels perceive their physical fitness as significantly lower across all fitness domains, including cardiorespiratory endurance, muscular strength, speed–agility, and flexibility. These results align with prior research suggesting that psychological distress, particularly anxiety, can negatively impact self-efficacy and perceived physical capacity in cancer survivors [53]. The interplay between anxiety and self-perception may be explained by cognitive distortions common in anxious individuals, leading to underestimation of their physical abilities despite a possible comparable objective performance [54]. Given that self-perceived physical fitness has been identified as a predictor of mortality and serves as a stable indicator of habitual PA levels [55,56], its evaluation may be as relevant as objective assessments in understanding long-term health risks. Moreover, although not specifically in LTBCSs, previous studies have reported strong correlations between self-reported and objectively measured fitness, reinforcing the validity of self-perception as a valuable tool for identifying individuals with lower physical ability and guiding tailored interventions [55,56]. Therefore, considering the possible impact of anxiety on both objectively measured and self-perceived physical fitness could provide a more comprehensive approach in the design of rehabilitation programs, allowing for more personalized and effective interventions in this population.

Although no statistically significant differences in overall MLTPA scores were found, a higher proportion of physically inactive LTBCSs was observed among those with elevated anxiety levels. This trend is consistent with previous studies indicating that anxiety can act as a barrier to engaging in regular PA, potentially due to fear of discomfort, avoidance behaviors, and fatigue-related concerns [10]. Given the well-documented benefits of PA in mitigating anxiety and other cancer-related symptoms [57], targeted interventions focusing on behavioral activation and structured exercise programs may be beneficial in this population. Future research should explore whether personalized psychological and PA interventions can enhance both perceived and actual physical fitness in anxious LTBCSs.

### 4.5. Health-Related Quality of Life

When it comes to HRQoL using the QLQ-C30, our LTBCSs with high anxiety levels exhibited notably lower scores in different functioning domains, as well as in “global health status” and in the “summary score”. Additionally, these LTBCSs reported significantly higher scores across all symptom scales and single-item measures, with the exception of “constipation”, when compared to LTBCSs with low anxiety levels. These findings are consistent with previous research indicating that anxiety can adversely affect physiological function, treatment adherence, psychological well-being, and overall HRQoL in BC patients and survivors [16,58].

Moreover, significant differences were observed in the BR23 module. LTBCSs experiencing high anxiety levels reported significantly lower scores in “body image” and “sexual functioning”, as well as higher scores in “future perspective” and all symptom scales, compared to their counterparts with low anxiety levels. These results align with a study which highlights that psychological distress, including anxiety, is prevalent among BC survivors and can profoundly impact various aspects of HRQoL, such as body image perception and satisfaction, sexual libido, increased concerns about the future, or other possible physical symptoms [59]. These findings underscore the critical need for comprehensive psychosocial interventions aimed at addressing anxiety and its associated effects on HRQoL in LTBCSs. Implementing routine distress screenings and providing tailored psychological support could be instrumental in enhancing the overall well-being and HRQoL for this population.

### 4.6. Correlation Analysis and Multiple Regression Analysis

The correlation analysis revealed significant positive associations between anxiety levels and various physical and psychological symptoms, including CRF, pain, dyspnea, insomnia, appetite loss, and financial difficulties. Conversely, we observed significant negative correlations between anxiety levels and factors such as role, emotional, cognitive, social, and sexual functioning, global health status, and self-perceived physical fitness. These findings align with the limited existing studies that have specifically conducted correlation analyses between anxiety and potential predictors in LTBCSs [6,8,60]. To date, most research has focused on descriptive comparisons, analyzing how different variables present depending on anxiety levels, rather than exploring their interrelationships and predictive value. However, our results highlight the importance of addressing these associations, as anxiety in LTBCSs is often linked to a greater symptom burden, including persistent CRF and chronic pain, which can exacerbate distress and impair daily functioning [6,8,60]. Given the scarcity of studies examining these correlations in long-term survivorship, further research is needed to investigate the potential interactions between anxiety and these physical and psychological factors beyond five years after a BC diagnosis.

Finally, our regression analysis revealed that total CRF (from the PFS) and cognitive functioning (from the QLQ-C30) were significant predictors of elevated anxiety levels among LTBCSs, collectively accounting for 47.0% of the variance. These findings suggest that higher CRF levels may increase anxiety, while impaired cognitive functioning also plays a role in heightened anxiety among LTBCSs. This finding aligns with existing literature that identifies various factors influencing anxiety in LTBCSs, though differences in study designs and reported predictors limit direct comparisons [5,6].

For instance, Breidenbach et al. (2022) found that younger age, lower income, and the presence of comorbidities significantly predicted higher anxiety levels in LTBCSs, explaining approximately 20% of the variance at follow-up [6]. Compared to our findings, which account for a larger proportion of variance (47.0%), this suggests that CRF and cognitive functioning may play an even greater role in long-term anxiety than previously recognized, warranting further investigation into their underlying mechanisms. Similarly, Cheng et al. (2021) [5] highlighted the role of coping profiles in predicting long-term anxiety trajectories in BC survivors, particularly emphasizing that maladaptive coping strategies were linked to heightened anxiety levels. Although the study did not report the variance explained by these predictors, its findings reinforce the relevance of psychological and behavioral factors in anxiety regulation [5]. Overall, while previous research has identified various contributors to anxiety in LTBCSs, our study provides novel insight into the strong predictive value of CRF and cognitive functioning. Hence, the 47.0% variance explained underscores the importance of integrating both physical and cognitive assessments in survivorship care, helping to identify individuals at risk and develop more targeted interventions.

### 4.7. Limitations and Strengths

This study has several limitations that should be acknowledged. First, although the VAS-A provided a rapid and practical method for stratifying anxiety levels, the use of a single-item tool rather than a standardized multidimensional instrument (e.g., HADS, STAI, and the Generalized Anxiety Disorder-7 (GAD-7)) may limit the depth of the anxiety assessment. Nonetheless, the VAS-A has been shown to correlate with more comprehensive scales in diverse populations and may serve as an efficient preliminary screening tool, particularly in clinical settings where time constraints preclude the use of lengthier instruments [25,26]. Moreover, we acknowledge that while our use of the VAS-A is grounded in previous work, it does not replace the depth of standardized multidimensional tools, and future studies should aim to validate these cut-offs further in specific cancer survivor populations. Second, the exclusive inclusion of female and Spanish-speaking LTBCSs limits the generalizability of findings to male survivors and non-Spanish-speaking or ethnically diverse groups. Third, the lack of information regarding pre-existing anxiety levels prior to cancer diagnosis restricts the ability to determine whether the observed anxiety is a result of the cancer experience or a pre-existing condition. Lastly, the cross-sectional nature of the study limits causal inferences, and although the cut-off points used were based on prior research in oncological populations [25], different thresholds might yield alternative results. Longitudinal studies are needed to clarify the directionality of these associations.

Despite its limitations, this study offers novel and clinically relevant insights into the relationship between anxiety and a range of health-related factors—physical, emotional, cognitive, and HRQoL domains—in LTBCSs. To our knowledge, it is one of the few studies to adopt a multidimensional approach that integrates CRF, pain, and self-perceived physical fitness as potential correlates of anxiety in long-term survivorship. The use of previously validated instruments in oncological populations strengthens the methodological robustness of the findings [24,28,29,30,34,41,42], and the regression model—explaining 47.0% of the variance in anxiety—underscores the clinical importance of these associations. Together, these results contribute to the scientific understanding of survivorship care and identify potential targets for individualized interventions aimed at improving both psychological and physical outcomes in LTBCSs.

## 5. Conclusions

In conclusion, nearly half of LTBCSs (46.25%) experienced high anxiety levels, which were linked to greater impairments in mood, self-perceived physical fitness, HRQoL, CRF, and pain. Total CRF and cognitive functioning significantly predicted anxiety, explaining 47.0% of its variance even five or more years post-diagnosis.

These results underscore the critical interaction between anxiety and key physical and psychological factors in long-term survivorship. Clinically, routine assessment of anxiety, CRF, and cognitive function in LTBCSs is essential for early identification of those at risk. Implementing targeted interventions—such as cognitive–behavioral therapy and exercise programs—could reduce anxiety symptoms and improve overall well-being. Future research should prioritize evaluating multidisciplinary interventions to optimize long-term health outcomes and HRQoL in this population.

## Figures and Tables

**Figure 1 life-15-00932-f001:**
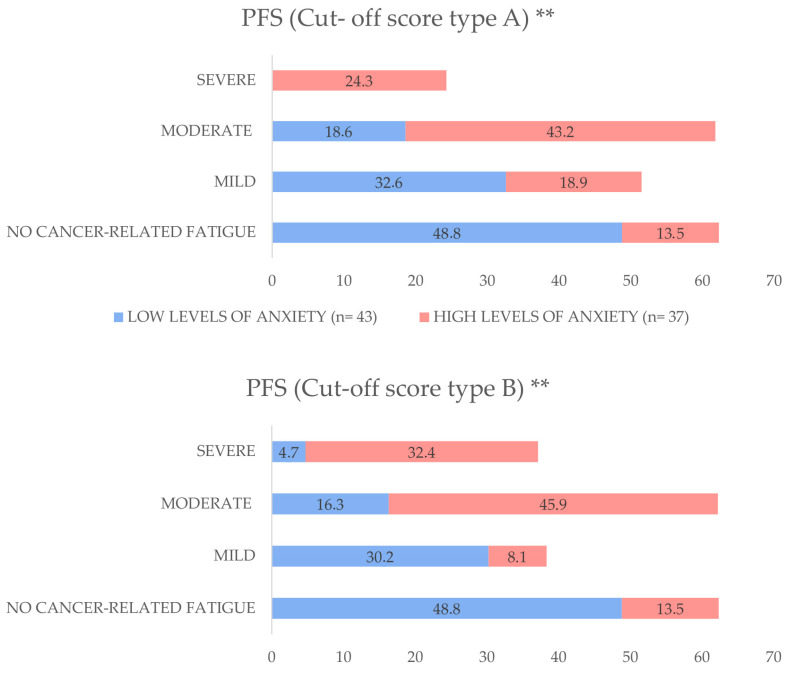
Distribution of cancer-related fatigue (CRF) severity by anxiety levels among long-term breast cancer survivors, expressed as percentages. Participants were stratified into low anxiety levels (VAS-A ≤ 3.4) and high anxiety levels (VAS-A ≥ 3.5). Fatigue severity was categorized using two cut-off score systems: Cut-off score A (No Fatigue: 0; Mild: 1–3; Moderate: 4–6; Severe: 7–10) and Cut-off score B (No Fatigue: 0; Mild: 1–2; Moderate: 3–5; Severe: 6–10). Between-group comparisons were performed using the Mann–Whitney U test. Abbreviations: PFS: Piper Fatigue Scale, VAS-A: Visual Analogue Scale for Anxiety, *n*: sample size. ** *p* < 0.01.

**Figure 2 life-15-00932-f002:**
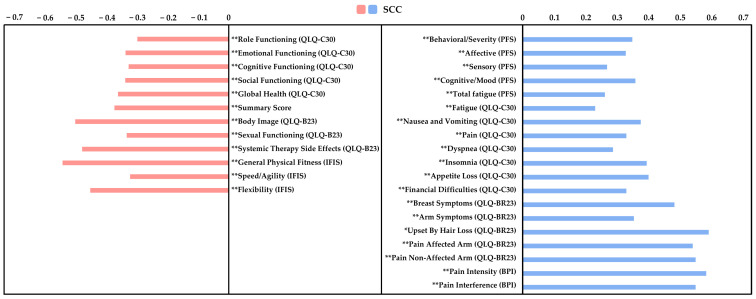
Spearman’s correlation coefficients between anxiety levels and multiple health-related variables in long-term breast cancer survivors. Anxiety was assessed using the Visual Analogue Scale for Anxiety (VAS-A), analyzed as a continuous (non-categorized) variable. The figure displays the strength and direction of associations with measures of fatigue, pain, self-perceived fitness, and health-related quality of life. Abbreviations: VAS-A: Visual Analogue Scale for Anxiety, PFS: Piper Fatigue Scale, BPI: Brief Pain Inventory, IFIS: International Fitness Scale, QLQ-C30: EORTC Core Quality of Life Questionnaire, QLQ-BR23: Breast Cancer-Specific Module, SCC: Spearman’s Correlation Coefficient. Significant correlation at * *p* < 0.05; ** *p* < 0.01.

**Table 1 life-15-00932-t001:** Demographic, clinical, and medical characteristics of the groups.

Characteristics	LTBCSs’ Anxiety Levels		*p*/*χ*^2^
	Low Levels of Anxiety	High Levels of Anxiety	
	≤3.4 (VAS-A) (*n* = 43)	≥3.5 (VAS-A) (*n* = 37)	
Mean age ± SD, years	48.41 ± 8.05	50.51 ± 8.02	0.63 ^a^
Mean time since diagnosis ± SD, months	91.52 ± 27.94	89.24 ± 30.99	0.49 ^a^
Mean time since the first surgery ± SD, months	88.43 ± 28.58	86.64 ± 31.23	0.50 ^a^
Marital status, *n* (%)			
Unmarried	6 (14)	7 (18.9)	
Married	33 (76.7)	22 (59.5)	
Divorced or separated	2 (4.7)	6 (16.2)	
Widowed	2 (4.7)	2 (5.4)	0.27 ^b^
Educational level, *n* (%)			
Primary school	15 (34.9)	19 (51.4)	
Secondary school	9 (20.9)	9 (24.3)	
University	19 (44.2)	9 (24.3)	0.16 ^b^
Employment status, *n* (%)			
Housewife	15 (34.9)	12 (27)	
Currently working	13 (30.2)	4 (10.8)	
Sick leave	11 (25.6)	15 (51.4)	
Retired	4 (9.3)	5 (10.8)	0.06 ^b^
Tobacco consumption, *n* (%)			
Non-consumption	24 (55.8)	16 (43.2)	
Smoker	9 (20.9)	10 (27)	
Ex-smoker	10 (23.3)	11 (29.7)	0.53 ^b^
Alcohol consumption, *n* (%)			
Non-consumption	13 (30.2)	17 (45.9)	
Monthly	7 (16.3)	12 (32.4)	
Weekly	20 (46.5)	7 (18.9)	
Daily	3 (7)	1 (2.7)	0.06 ^b^
Menopause, *n* (%)			
No	6 (14)	5 (13.5)	
Yes	37 (86)	32 (86.5)	0.95 ^b^
Family history of breast cancer, *n* (%)			
No	25 (58.1)	14 (37.8)	
Yes	18 (41.9)	23 (62.2)	0.07 ^b^
Tumor stage, *n* (%)			
I	14 (32.6)	9 (24.3)	
II	23 (53.5)	23 (62.2)	
IIIa	6 (14.0)	5 (13.5)	0.69 ^b^
Type of treatment, *n* (%)			
None	0 (0)	0 (0)	
Radiotherapy	0 (0)	3 (8.1)	
Chemotherapy	2 (4.7)	4 (10.8)	
Radiotherapy and chemotherapy	41 (95.3)	30 (81.1)	0.08 ^b^
Surgery, *n* (%)			
Lumpectomy	8 (18.6)	8 (21.6)	
Quadrantectomy	26 (60.5)	13 (35.1)	
Unilateral mastectomy	9 (20.9)	12 (32.4)	
Bilateral mastectomy	0 (0)	4 (10.8)	0.06 ^b^
Type of medication, *n* (%)			
None	11 (25.6)	8 (21.6)	
Tamoxifen	17 (39.5)	14 (37.8)	
Other types	15 (34.9)	15 (40.5)	0.85 ^b^
Metastasis, *n* (%)			
No	36 (83.7)	30 (81.1)	
Yes	7 (16.3)	7 (18.9)	0.75 ^b^
Recurrence, *n* (%)			
No	36 (83.7)	31 (83.8)	
Yes	7 (16.3)	6 (16.2)	0.99 ^b^
Currently seeing a psychologist or in the last three months, *n* (%)			
No	16 (37.2)	17 (45.9)	
Yes	27 (62.8)	20 (54.1)	0.42 ^b^
Currently seeing a physiotherapist or in the last three months, *n* (%)			
No	17 (39.5)	12 (32.4)	
Yes	26 (60.5)	25 (67.6)	0.51 ^b^

Abbreviations: LTBCSs: Long-term breast cancer survivors, VAS-A: Visual Analogue Scale for Anxiety, *n*: Sample size, SD: Standard deviation. *p* values for between-group differences were calculated using *t*-test ^a^ and χ^2^ for categorical variables ^b^.

**Table 2 life-15-00932-t002:** Mood state, cancer-related fatigue, pain, self-perceived physical fitness, and physical activity level values between groups.

Variables	LTBCSs’ Anxiety Levels		*p*/*χ*^2^	Cohen’s *d*
	Low Levels of Anxiety	High Levels of Anxiety		
	≤3.4 (VAS-A) (*n* = 43)	≥3.5 (VAS-A) (*n* = 37)		
EVEA, mean ± SD (95% CI) ^a^				
Sadness–Depression	1.10 ± 1.38 (0.67–1.53)	5.02 ± 2.30 (4.25–5.79)	<0.01 **	>1.20
Anger–Hostility	0.64 ± 0.88 (0.37–0.91)	4.11 ± 2.46 (3.29–4.93)	<0.01 **	>1.20
Happiness	6.19 ± 2.41 (5.44–6.93)	5.76 ± 8.55 (2.91–8.61)	<0.01 **	0.06
PFS (domains), mean ± SD (95% CI) ^a^				
Behavioral/Severity	1.63 ± 2.07 (0.99–2.26)	4.57 ± 2.86 (3.62–5.53)	<0.01 **	1.18
Affective	1.77 ± 2.35 (1.05–2.50)	5.11 ± 3.07 (4.08–6.13)	<0.01 **	>1.20
Sensory	2.08 ± 2.50 (1.31–2.85)	5.09 ± 2.91 (4.12–6.06)	<0.01 **	1.11
Cognitive/Mood	1.72 ± 2.02 (1.09–2.34)	4.92 ± 2.93 (3.94–5.90)	<0.01 **	>1.20
Total	1.79 ± 2.03 (1.16–2.41)	4.94 ± 2.68 (4.04–5.83)	<0.01 **	>1.20
VAS (cm), mean ± SD (95% CI) ^a^				
Dominant arm	1.23 ± 1.64 (0.72–1.73)	3.56 ± 2.92 (2.59–4.54)	<0.01 **	0.98
Non-dominant arm	0.32 ± 0.89 (0.05–0.60)	2.78 ± 3.39 (1.65–3.91)	<0.01 **	0.99
BPI, mean ± SD (95% CI) ^a^				
Intensity	1.48 ± 1.66 (0.97–1.99)	3.19 ± 2.97 (2.20–4.19)	0.01 *	0.71
Interference	1.01 ± 1.66 (0.50–1.52)	3.20 ± 3.10 (2.17– 4.24)	0.01 *	0.88
IFIS, mean ± SD (95% CI)				
General physical fitness	3.55 ± 0.98 (3.25–3.86)	2.97 ± 0.92 (2.66–3.28)	<0.01 **	0.61
Cardiorespiratory fitness	2.95 ± 1.23 (2.57–3.33)	2.81 ± 0.93 (2.49–3.12)	0.05 *	0.13
Muscular strength	3.13 ± 1.05 (2.81–3.46)	2.56 ± 0.89 (2.26–2.86)	0.01 *	0.59
Speed/Agility	3.25 ± 0.95 (2.96–3.54)	2.59 ± 0.86 (2.30–2.88)	<0.01 **	0.73
Flexibility	3.09 ± 0.94 (2.80–3.38)	2.62 ± 1.08 (2.25–2.98)	0.03 *	0.46
MLTPA, *n* (%) ^b^				
Inactive: ≤3 (MET—hour/week)	8 (18.6)	13 (35.1)	0.24	
Low active: 3.1–7.4 (MET—hour/week)	19 (44.2)	13 (35.1)		
Active: ≥7.5 (MET—hour/week)	16 (37.2)	11 (29.7)		-

Abbreviations: LTBCSs: Long-term Breast Cancer Survivors, VAS: Visual Analogue Scale, VAS-A: Visual Analogue Scale for Anxiety, EVEA: Scale for Mood Assessment, PFS: Piper Fatigue Scale, BPI: Brief Pain Inventory, IFIS: International Fitness Scale, MLTPA: Minnesota Leisure Time Physical Activity, MET: Metabolic Equivalent of Task, CI: Confidence interval, *n*: Sample size, SD: Standard deviation. *p* values for between-group differences were calculated using the Mann–Whitney U test for continuous variables ^a^ and *χ*^2^ for categorical variables ^b^. Between-group effect sizes were calculated using Cohen’s *d* for continuous variables ^a^. Note: The row corresponding to Anxiety (EVEA) has not been included in the table nor in the main analysis, as it was primarily assessed using the Visual Analogue Scale for Anxiety (VAS-A). * *p* ˂ 0.05. ** *p* < 0.01.

**Table 3 life-15-00932-t003:** Health-related quality of life values between groups.

Variables	LTBCSs’ Anxiety Levels		*p* Values	Cohen’s *d*
	Low Levels of Anxiety	High Levels of Anxiety		
	≤3.4 (VAS-A) (*n* = 43)	≥3.5 (VAS-A) (*n* = 37)		
Functioning Scales QLQ-C30, mean ± SD (95% CI)				
Physical Functioning	29.06 ± 16.70 (23.92–34.20)	36.02 ± 21.34 (28.90–43.14)	0.20	0.36
Role Functioning	90.10 ± 14.78 (85.55–94.65)	71.72 ± 21.68 (64.49–78.95)	<0.01 **	0.99
Emotional Functioning	88.37 ± 23.43 (81.15–95.58)	68.91 ± 32.19 (58.18–79.65)	<0.01 **	0.69
Cognitive Functioning	79.06 ± 21.15 (72.55–85.58)	48.64 ± 32.30 (37.87–59.41)	<0.01 **	1.11
Social Functioning	74.41 ± 24.21 (66.96–81.87)	47.29 ± 31.79 (36.69–57.89)	<0.01 **	0.96
Symptom Scales QLQ-C30, mean ± SD (95% CI)				
Fatigue	60.36 ± 36.50 (48.18–72.53)	82.94 ± 23.14 (75.82–90.06)	<0.01 **	0.74
Nausea and Vomiting	22.99 ± 24.41 (15.48–30.51)	52.25 ± 31.30 (41.81–62.69)	<0.01 **	1.04
Pain	3.48 ± 11.81 (−0.14–7.12)	12.16 ± 22.78 (4.56–19.75)	0.02 *	0.48
Single Items QLQ-C30, mean ± SD (95% CI)				
Dyspnea	25.58 ± 23.94 (18.21–32.95)	54.05 ± 34.11 (42.68–65.42)	<0.01 **	0.97
Insomnia	13.95 ± 25.44 (6.12–21.78)	34.23 ± 33.78 (22.97–45.49)	<0.01 **	0.68
Appetite Loss	42.24 ± 32.19 (32.34–52.15)	59.45 ± 34.36 (48.00–70.91)	0.02 *	0.52
Constipation	9.30 ± 23.37 (2.10–16.49)	14.41 ± 27.82 (5.13–23.69)	0.34	0.20
Diarrhea	18.21 ± 27.41 (9.77–26.65)	33.33 ± 35.13 (21.61–45.04)	0.04 *	0.48
Financial Difficulties	4.65 ± 11.68 (1.05–8.24)	20.72 ± 29.76 (10.79–30.64)	0.04 *	0.71
Global Health Status QLQ-C30, mean ± SD (95% CI)				
Global Health Status	35.22 ± 39.16 (22.16–48.28)	12.40 ± 24.15 (4.97–19.83)	<0.01 **	0.70
Summary Score QLQ-C30, mean ± SD (95% CI)				
Summary Score	72.48 ± 11.22 (69.02–75.93)	57.87 ± 16.25 (52.45–63.29)	<0.01 **	1.05
Functional Scales QLQ-BR23, mean ± SD (95% CI)				
Body Image	70.93 ± 19.36 (64.97–76.88)	55.18 ± 23.27 (47.42–62.93)	<0.01 **	0.74
Sexual Functioning	85.27 ± 19.65 (79.22–91.32)	67.79 ± 33.22 (56.71–78.87)	0.01 *	0.64
Sexual Enjoyment	23.64 ± 19.66 (17.59–29.69)	19.82 ± 22.85 (12.20–27.43)	0.23	0.18
Future Perspective	26.12 ± 16.21 (17.38–34.86)	37.20 ± 23.24 (30.05–44.36)	0.01 *	0.55
Symptom Scales QLQ-BR23, mean ± SD (95% CI)				
Systemic Therapy Side Effects	42.34 ± 37.39 (29.87–54.80)	65.11 ± 36.33 (53.93–76.29)	<0.01 **	0.62
Breast Symptoms	20.01 ± 18.33 (14.37–25.65)	38.60 ± 22.88 (30.97–46.24)	<0.01 **	0.90
Arm Symptoms	18.02 ± 20.32 (11.76–24.27)	35.58 ± 31.52 (25.07–46.09)	0.01 *	0.66
Upset By Hair Loss	22.47 ± 23.44 (15.26–29.69)	42.64 ± 34.99 (30.97–54.30)	0.01 *	0.68

Abbreviations: LTBCSs: Long-term Breast Cancer Survivors, VAS-A: Visual Analogue Scale for Anxiety, QLQ-C30: The EORTC Core Quality of Life Quality of Life Questionnaire, QLQ-BR23: The Breast Cancer-Specific Module, CI: Confidence interval, *n*: Sample size, SD: Standard deviation. *p* values for between-group differences were calculated using the Mann–Whitney U test. Between-group effect sizes were calculated using Cohen’s *d*. Significant difference at * *p* ˂ 0.05. ** *p* < 0.01.

**Table 4 life-15-00932-t004:** Summary of Stepwise Multiple Regression Analysis to determine predictors of anxiety using the Visual Analogue Scale for Anxiety (VAS-A).

Model	Variables/Predictors	β	95% CI	t	*p* Values	Linear Regression Equation Y = a + bX
**Model 1** **(r^2^ = 0.432)**	Total CRF (PFS)	0.65	0.45 ± 0.76	7.69	<0.01 **	Anxiety = 1.08 + (0.60 Total CRF)
**Model 2** **(r^2^ = 0.470)**	Total CRF (PFS)	0.51	0.28 ± 0.66	4.97	<0.01 **	Anxiety = 2.86 + (0.47 Total CRF) + (−0.02 Cognitive Functioning)
	Cognitive Functioning (QLQ-C30)	−0.24	−0.03 ± −0.00	−2.37	0.02 *	

Dependent variable: Anxiety (VAS-A), *r*^2^: Adjusted coefficient of determination, *β*: Regression coefficient, *t*: Coefficient *t*-value. Abbreviations: VAS-A: Visual Analogue Scale for Anxiety, CRF: Cancer-Related Fatigue, PFS: Piper Fatigue Scale, QLQC30: The EORTC Core Quality of Life Quality of Life Questionnaire, CI: Confidence Interval. Note: For this analysis, the VAS-A data as the dependent variable was used in its non-categorized version. Significant difference at * *p* ˂ 0.05. ** *p* < 0.01.

## Data Availability

The datasets generated during and/or analyzed during the current study are available from the corresponding author on reasonable request.

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
