# Peer review of "Influence and Predictors of Anxiety on Health Status ≥ 5 Years Beyond Breast Cancer Diagnosis in Spain: A Cross-Sectional Study"

_life, 2025, doi:10.3390/life15060932_

Round 1

Reviewer 1 Report

Comments and Suggestions for Authors

The manuscript addresses an important and clinically relevant topic regarding anxiety among long-term breast cancer survivors (LTBCSs). The overall study design, methodology, and data analysis are adequate, and the findings contribute valuable information to survivorship care. However, the manuscript needs several improvements before it can be considered for publication.

Major Concerns

Title Formatting

Title capitalization is inconsistent. It should be: "Influence and Predictors of Anxiety on Health Status ≥5 Years Beyond Breast Cancer Diagnosis: A Cross-Sectional Study"

Abstract Structure

The abstract is informative but overly dense.

Suggested improvements:

Separate the purpose, methods, results, and conclusion more clearly.

Avoid repeating results twice.

Add a clearer statement on clinical implications.

Example correction: "Purpose: To explore the relationship between anxiety and health outcomes in LTBCSs, and identify predictors of anxiety."

Introduction

Too lengthy and redundant: The Introduction is well-written but repeats similar ideas (e.g., the distinction between anxiety and depression) several times.

Suggested improvements:

Shorten background paragraphs.

Emphasize the gap in previous studies earlier and more crisply.

Methods

Ethical Approval: Stated correctly, but the phrasing could be clearer and moved earlier in the Methods section for better flow.

VAS-A for Anxiety:

No clear validation of the adapted Visual Analogue Scale for Anxiety (VAS-A) is given for this specific population.

Need a brief justification why a simple VAS was used over standardized anxiety questionnaires (e.g., GAD-7, HADS-Anxiety).

Group Division Cut-offs:

The division of anxiety groups at 3.4/3.5 cutoff is mentioned, but the rationale is not strong. Clarify the clinical relevance.

Sample Size Calculation:

The G*Power calculation is included, but it would be better to mention the expected effect size from prior studies if available.

Statistical Analysis

Multiple Testing Correction:

Given the large number of comparisons (many variables across groups), it would be appropriate to mention whether any correction for multiple testing was applied (e.g., Bonferroni, FDR).

Otherwise, readers might question the risk of Type I error (false positives).

Results

Overloaded Tables:

Tables 1–3 are too crowded and difficult to read.

Suggested improvements:

Split into sub-tables or focus only on significant variables in the main text.

Move full demographic tables to Supplementary Material if allowed by journal guidelines.

Figures:

Figure 1 and Figure 2 captions need to be more descriptive (should stand alone without referring to text).

Improve graphical quality (figures are a bit low-resolution in the file).

Discussion

Well-organized but verbose:

Good integration of findings and literature.

However, several paragraphs are too repetitive (e.g., discussion on CRF and anxiety).

Suggested improvements:

Summarize findings in a more concise way.

Strengthen the "clinical implication" and "future research" sections.

Causal Language Warning:

Given the cross-sectional design, be careful not to imply causality ("Anxiety leads to worse outcomes"). Use language like "associated with" or "linked to".

Limitations

Good that limitations are acknowledged, but should also discuss:

Lack of pre-cancer anxiety baseline.

Limited generalizability to other ethnicities (only Spanish-speaking participants).

References

Reference style is inconsistent (spacing errors, line breaks).

Carefully reformat based on journal instructions.

Minor Concerns

Typographical Errors:

Several sentences have double spaces between words.

Some inconsistent use of periods after abbreviations (e.g., "CRF" vs "CRF.").

Redundant phrases like "the level of anxiety level" — needs to be corrected to just "anxiety level".

Comments on the Quality of English Language

Language Polishing:

Use more active voice instead of passive. Example: "Assessments were conducted" → "The research team conducted assessments."

Reword awkward sentences: "Following the same protocol as hospital-referred participants, assessments were conducted at the association's premises."

→ "The same protocol used for hospital-referred participants was applied for assessments at association premises."

Author Response

Clara Pujol Fuentes

E-mail address

clara.pujol@universidadeuropea.es

+34 661075990
Full postal address

FIBIO Research Group,

Department of Physiotherapy,
Faculty of Health Sciences, European University of Valencia,
46010, Valencia, Spain

Editorial Reviewer 1

Life

2 June 2025

Dear Reviewer 1,

Please find below the answers to each of your contributions

Comments and Suggestions for Authors

“The manuscript addresses an important and clinically relevant topic regarding anxiety among long-term breast cancer survivors (LTBCSs). The overall study design, methodology, and data analysis are adequate, and the findings contribute valuable information to survivorship care. However, the manuscript needs several improvements before it can be considered for publication”.

Author response:

First of all, we would like to thank you for your words and the time dedicated to the understanding and improvement of this scientific work. In this way, and from here on, all the answers are detailed, individually and by sections, to each of your suggestions or comments.

With this in mind, we believe that, thanks to the reviewer's contributions and suggestions, together with all our responses, this scientific research is now much easier to read and understand.

Tittle formatting

  1. Reviewer comment: “Title capitalization is inconsistent. It should be: "Influence and Predictors of Anxiety on Health Status ≥5 Years Beyond Breast Cancer Diagnosis: A Cross-Sectional Study".

Author response:

Thank you for your observation. We have corrected the title to ensure consistent capitalization as suggested:

“Influence and Predictors of Anxiety on Health Status ≥ 5 Years Beyond Breast Cancer Diagnosis in Spain: A Cross-Sectional Study.”

Abstract structure

  1. Reviewer comment: The abstract is informative but overly dense. Suggested improvements: Separate the purpose, methods, results, and conclusion more clearly. Avoid repeating results twice. Add a clearer statement on clinical implications.

Example correction: "Purpose: To explore the relationship between anxiety and health outcomes in LTBCSs, and identify predictors of anxiety".

Author response:

Thank you very much for this valuable suggestion. We have revised the abstract to enhance clarity and readability by explicitly separating the sections (Purpose, Methods, Results, and Conclusion), as recommended. We have also streamlined the content to avoid redundancy, particularly in the Results and Conclusion sections, and rephrased the clinical implications to make them more explicit and direct. We believe the revised version is now more concise, better structured, and more accessible to the reader, while preserving all essential scientific content. We appreciate your insightful feedback, which has helped us improve the quality of the manuscript (see lines 20 to 22).

“Abstract

Purpose: To explore the association between anxiety levels and health outcomes in long-term breast cancer survivors (LTBCSs), and to identify predictors of anxiety in this population.

Methods: A multicenter cross-sectional study was conducted with 80 LTBCSs, categorized into two groups based on their anxiety levels: low anxiety (≤ 3.4) and high anxiety (≥ 3.5). The analysis focused on variables assessed at least five years after diagnosis, including sociodemographic and clinical data, mood, cancer-related fatigue (CRF), pain, self-perceived physical fitness, physical activity (PA), and health-related quality of life (HRQoL). ANOVA, Mann-Whitney U, and Chi-square tests were conducted, along with correlation and multiple regression analysis. Effect sizes were calculated using Cohen's d.

Results: Among the participants, 46.25% exhibited higher anxiety levels. This group showed significantly worse mood, self-perceived physical fitness, HRQoL, and elevated CRF and pain (p < 0.05). Regression analysis identified “total CRF” (β = 0.51; p < 0.01) and “cognitive functioning” (β = -0.24; p = 0.02) as significant predictors of higher levels of anxiety (r²adjusted = .470).

Conclusion: Anxiety significantly impacts multiple dimensions of health in LTBCSs. Total CRF and cognitive functioning are key predictors of anxiety. These findings have direct clinical implications: routine psychological and physical assessments should be integrated into survivorship care to identify individuals at risk and inform targeted interventions to enhance long-term well-being and HRQoL”.

Introduction

  1. Reviewer comment: “Too lengthy and redundant: The Introduction is well-written but repeats similar ideas (e.g., the distinction between anxiety and depression) several times.

Suggested improvements: Shorten background paragraphs. Emphasize the gap in previous studies earlier and more crisply”.

Author response:

Thank you for your thoughtful feedback regarding the Introduction. In response, we have significantly revised this section to improve its clarity and conciseness. Specifically, we removed several redundant sentences from the second paragraph, including four references, and substantially restructured the third paragraph to avoid overlap and to highlight the research gap more directly. Additionally, we refined the concluding sentences to improve the logical flow and ensure alignment with the aims stated in the abstract. We maintained the first and second paragraphs largely unchanged, as we believe they provide a concise and relevant background that supports the rationale of the study. As a result of these edits, the length of the Introduction has been reduced from 459 to 389 words. We appreciate your suggestion, which has helped us improve the precision and coherence of this section (see lines 73 to 80 as well as lines 82 to 87).

“Nevertheless, many existing studies do not provide an integrative analysis of factors associated with anxiety in LTBCSs, which limits the design of targeted interventions. Methodological heterogeneity and inconsistent outcome measures further hinder the interpretation and generalization of findings [5,15–17]. A clearer understanding of anxiety and its predictors in this population is therefore essential to develop personalized, multidisciplinary care strategies. Recognizing anxiety as a distinct clinical concern may help improve long-term outcomes, as it remains frequently underdiagnosed despite its substantial impact on HRQoL and functional recovery”.

“Considering the aforementioned, the present study aims to (1) explore the association between anxiety levels and health outcomes in LTBCSs, and (2) identify predictors of anxiety in this population ≥5 years beyond BC diagnosis. By focusing solely on anxiety, this study seeks to fill a critical gap in survivorship research and support the development of more individualized interventions aimed to improve the long-term emotional and physical well-being of LTBCSs”.

Methods

  1. Reviewer comment: “Ethical Approval: Stated correctly, but the phrasing could be clearer and moved earlier in the Methods section for better flow”.

Author response:

Thank you for your helpful comment. In response, we have restructured the relevant section by placing the ethical approval statement earlier, immediately after the study design. Additionally, we have divided the information into two separate paragraphs to improve readability and flow. We trust that these changes enhance the clarity and organization of the Methods section, in line with your recommendation (see lines 90 to 103).

“This study followed a cross-sectional and descriptive design and was conducted in accordance with the Declaration of Helsinki (version 14/2017) [18]. Ethical approval was obtained from the Biomedical Research Ethics Committee of Granada (CEIm) (Ref: 1038-N-16 I.P/07/26/2018). A total of 80 LTBCSs were recruited in 2024 through the oncology department of the San Cecilio University Hospital of Granada and various BC associations in Granada and Málaga.

Sample size was calculated using G*Power (Version 3.1.9.7), assuming a medium effect size (f = 0.25; Cohen’s d = 0.50), α = 0.05, and power = 0.80. This assumption is supported by studies on anxiety and related outcomes [19,20] and by similar sample size calculation approaches used in cancer survivor populations for other clinical variables [21]. A minimum of 72 participants was required, and 80 were ultimately recruited. Evaluations were conducted either at the Faculty of Health Sciences (University of Granada) or within the facilities of the collaborating associations”.

  1. Reviewer comment: “VAS-A for Anxiety: No clear validation of the adapted Visual Analogue Scale for Anxiety (VAS-A) is given for this specific population. Need a brief justification why a simple VAS was used over standardized anxiety questionnaires (e.g., GAD-7, HADS-Anxiety)”.

Author response:

We thank the reviewer for raising this relevant point. We acknowledge that widely used, multidimensional standardized anxiety questionnaires such as the GAD-7 or HADS-Anxiety provide a more comprehensive evaluation of anxiety symptoms. However, in the present study, we deliberately chose to use the Visual Analogue Scale for Anxiety (VAS-A) for pragmatic and clinical reasons.

In real-world clinical contexts—especially in survivorship care—time constraints often limit the feasibility of administering lengthy psychometric tools. In such scenarios, brief and easy-to-administer screening instruments like the VAS-A can serve as valuable initial tools to detect anxiety symptoms. As such, our goal was to identify survivors with potentially elevated anxiety levels who could then be referred for more detailed psychological evaluation using multidimensional measures. This approach aligns with current clinical practice where efficiency and feasibility are crucial.

Importantly, the VAS-A has demonstrated utility in various populations, including individuals with cancer. For example, Labaste et al. (2019) validated the VAS-A for assessing postoperative anxiety in a diverse cohort that included cancer patients. Their findings supported the VAS-A as a simple, reliable tool to detect the anxiety component of postoperative distress and to support decision-making regarding anxiety management (Nurs Open. 2019;6(4):1323–1330). Furthermore, this study was also the source of the cut-off points used in our work, thus providing an empirical basis for the stratification applied to our sample.

Moreover, Ducoulombier et al. (2020) confirmed the effectiveness of the VAS-A in hospitalized patients experiencing pain, identifying a clear anxiety threshold and supporting its role as a valid screening method in inpatient settings. The authors emphasized its simplicity, familiarity among healthcare professionals, and potential for integration into routine care (Pain Manag Nurs. 2020;21(6):572–578).

Similarly, Lavedán Santamaría et al. (2022) showed significant diagnostic concordance between the VAS-A and the State-Trait Anxiety Inventory (STAI) in nursing students during the COVID-19 pandemic. Their results indicated that the VAS-A could serve as a rapid diagnostic or pre-diagnostic tool during crises or time-sensitive situations (Int J Environ Res Public Health. 2022;19(12):7053).

In addition, Widyastuty et al. (2019) demonstrated a significant correlation between VAS scores and HADS-D scores in cervical cancer patients, reinforcing the tool’s potential to reflect psychological distress levels reliably (Open Access Maced J Med Sci. 2019;7(16):2634–2637).

Although we are aware that the VAS-A has not been specifically validated in long-term breast cancer survivors, it has been previously employed in studies involving oncological populations—such as patients with cervical cancer or individuals in postoperative cancer care settings—which supports its applicability and relevance in the context of cancer survivorship.

  • Labaste F, Ferré F, Combelles H, Rey V, Foissac JC, Senechal A, Conil JM, Minville V. Validation of a visual analogue scale for the evaluation of the postoperative anxiety: A prospective observational study. Nurs Open. 2019 Jul 11;6(4):1323-1330. doi: 10.1002/nop2.330. PMID: 31660159; PMCID: PMC6805714.

  • Ducoulombier V, Chiquet R, Graf S, Leroy B, Bouquet G, Verdun S, Martellier F, Versavel A, Kone A, Lacroix K, Duthoit D, Lenglet Q, Devaux A, Jeanson R, Lefebvre A, Coviaux B, Calais G, Grimbert A, Ledein M, Moukagni M, Pascart T, Houvenagel E. Usefulness of a Visual Analog Scale for Measuring Anxiety in Hospitalized Patients Experiencing Pain: A Multicenter Cross-Sectional Study. Pain Manag Nurs. 2020 Dec;21(6):572-578. doi: 10.1016/j.pmn.2020.03.004. Epub 2020 May 1. PMID: 32362472.

  • Lavedán Santamaría A, Masot O, Canet Velez O, Botigué T, Cemeli Sánchez T, Roca J. Diagnostic Concordance between the Visual Analogue Anxiety Scale (VAS-A) and the State-Trait Anxiety Inventory (STAI) in Nursing Students during the COVID-19 Pandemic. Int J Environ Res Public Health. 2022 Jun 9;19(12):7053. doi: 10.3390/ijerph19127053. PMID: 35742303; PMCID: PMC9222809.

  • Widyastuty A, Effendy E, Amin MM. Correlation between Visual Analogue Scale Score and Hospital Anxiety Depression Scale-Depression Score in Patients with Cervical Cancer in the Hospital Vina Cancer, Medan. Open Access Maced J Med Sci. 2019 Aug 20;7(16):2634-2637. doi: 10.3889/oamjms.2019.473. PMID: 31777622; PMCID: PMC6876805.

With all this in mind, and following the revisions, the new paragraph would read as follows (see lines 129 to 143):

“Anxiety levels were measured to classify participants into groups using an adapted version of the Visual Analogue Scale (VAS) for Anxiety (VAS-A). This scale consists of a 10 cm horizontal line ranging from "no anxiety" (0) on the left to "worst possible anxiety" (10) on the right. Although the concept of a VAS adapted to measure anxiety was proposed in 1976 and first applied in 1988 for patients undergoing dental treatment [22], it has since demonstrated utility in various clinical populations, including women with BC [23]. More recent studies have shown a significant correlation between VAS-A scores and those obtained with standardized instruments like the Hospital Anxiety and Depression Scale (HADS) or the State-Trait Anxiety Inventory (STAI), suggesting its validity as a rapid screening tool for anxiety [24,25]. As such, it may serve as an efficient preliminary screening tool for anxiety and potential anxiety symptoms, particularly in clinical contexts where time constraints preclude the use of more comprehensive measures. Therefore, and following similar recommendations for cut-off points [24], participants were classified into two groups: those with lower anxiety levels (≤ 3.4) and those with higher anxiety levels (≥ 3.5)”.

Additionally, in response to this comment, we have now included in the Limitations section a statement acknowledging that the use of the VAS-A does not allow for a multidimensional assessment of anxiety, which would have been possible with more comprehensive tools such as the HADS or GAD-7 (see lines 594 to 612):

“This study has several limitations that should be acknowledged. First, although the VAS-A provided a rapid and practical method for stratifying anxiety levels, the use of a single-item tool rather than a standardized multidimensional instrument (e.g., HADS, STAI, and the Generalized Anxiety Disorder-7 (GAD-7)) may limit the depth of the anxiety assessment. Nonetheless, the VAS-A has been shown to correlate with more comprehensive scales in diverse populations, and may serve as an efficient preliminary screening tool, particularly in clinical settings where time constraints preclude the use of lengthier instruments [24,25]. Moreover, we acknowledge that while our use of the VAS-A is grounded in previous work, it does not replace the depth of standardized multidimensional tools, and future studies should aim to validate these cut-offs further in specific cancer survivor populations. Second, the exclusive inclusion of female and Spanish-speaking LTBCSs limits the generalizability of findings to male survivors and non-Spanish-speaking or ethnically diverse groups. Third, the lack of information regarding pre-existing anxiety levels prior to cancer diagnosis restricts the ability to determine whether the observed anxiety is a result of the cancer experience or a pre-existing condition. Lastly, the cross-sectional nature of the study limits causal inferences, and although the cut-off points used were based on prior research in oncological populations [24], different thresholds might yield alternative results. Longitudinal studies are needed to clarify the directionality of these associations”.

We also considered it appropriate to revise the strengths section to enhance clarity, precision, and consistency with the study's findings (see lines 614 to 624):

“Despite its limitations, this study offers novel and clinically relevant insights into the relationship between anxiety and a range of health-related factors—physical, emotional, cognitive, and HRQoL domains—in LTBCSs. To our knowledge, it is one of the few studies to adopt a multidimensional approach that integrates CRF, pain, and self-perceived physical fitness as potential correlates of anxiety in long-term survivorship. The use of previously validated instruments in oncological populations strengthens the methodological robustness of the findings [23,27-29,33,40,41], and the regression model—explaining 47.0% of the variance in anxiety—underscores the clinical importance of these associations. Together, these results contribute to the scientific understanding of survivorship care and identify potential targets for individualized interventions aimed at improving both psychological and physical outcomes in LTBCSs”.

  1. Reviewer comment: “Group Division Cut-offs: The division of anxiety groups at 3.4/3.5 cutoff is mentioned, but the rationale is not strong. Clarify the clinical relevance”.

Author response:

We appreciate the reviewer’s comment regarding the need to clarify the rationale and clinical relevance behind the anxiety group cut-off points (≤3.4 / ≥3.5). The chosen thresholds were not arbitrarily selected, but rather based on prior evidence from Labaste et al. (2019), who validated the use of a Visual Analogue Scale for Anxiety (VAS-A) in a mixed postoperative population that included cancer patients. In their study, they identified a cut-off score of 34/100 (i.e., 3.4/10) as a meaningful threshold to detect clinically relevant anxiety, suggesting that patients with scores equal to or above this value should be closely monitored and potentially referred for further psychological evaluation.

Although the VAS-A is not a multidimensional tool like the GAD-7 or HADS-Anxiety, it has shown utility in several clinical populations—including oncology contexts—as a quick screening instrument. For instance, other studies have reported significant correlations between VAS-A scores and more comprehensive tools like the HADS or STAI (Widyastuty et al., 2019; Lavedán Santamaría et al., 2022), supporting its use as a practical alternative in time-constrained settings. Ducoulombier et al. (2020) also emphasized its clinical applicability and ease of use by healthcare professionals in routine hospital care, especially when rapid identification of anxiety is required to adapt care plans.

Based on this body of literature, we believe that using the 3.4/3.5 cut-off allows for a clinically meaningful stratification of anxiety levels, balancing the need for a practical approach with evidence-informed thresholds. Nonetheless, we have acknowledged in the limitations section that the use of a VAS-A—while grounded in previous work—does not replace the depth provided by standardized multidimensional assessments, and future studies should aim to validate these cut-offs further in specific cancer survivor populations (see lines 594 to 612):

“This study has several limitations that should be acknowledged. First, although the VAS-A provided a rapid and practical method for stratifying anxiety levels, the use of a single-item tool rather than a standardized multidimensional instrument (e.g., HADS, STAI, and the Generalized Anxiety Disorder-7 (GAD-7)) may limit the depth of the anxiety assessment. Nonetheless, the VAS-A has been shown to correlate with more comprehensive scales in diverse populations, and may serve as an efficient preliminary screening tool, particularly in clinical settings where time constraints preclude the use of lengthier instruments [24,25]. Moreover, we acknowledge that while our use of the VAS-A is grounded in previous work, it does not replace the depth of standardized multidimensional tools, and future studies should aim to validate these cut-offs further in specific cancer survivor populations. Second, the exclusive inclusion of female and Spanish-speaking LTBCSs limits the generalizability of findings to male survivors and non-Spanish-speaking or ethnically diverse groups. Third, the lack of information regarding pre-existing anxiety levels prior to cancer diagnosis restricts the ability to determine whether the observed anxiety is a result of the cancer experience or a pre-existing condition. Lastly, the cross-sectional nature of the study limits causal inferences, and although the cut-off points used were based on prior research in oncological populations [24], different thresholds might yield alternative results. Longitudinal studies are needed to clarify the directionality of these associations”.

  1. Reviewer comment: “Sample Size Calculation: The G*Power calculation is included, but it would be better to mention the expected effect size from prior studies if available”.

Author response:

Thank you for your valuable comment regarding the sample size calculation.
The sample size was calculated using G*Power software (Version 3.1.9.7), assuming a medium effect size (f = 0.25, corresponding to Cohen’s d = 0.50), with a significance level of 0.05 and power of 80%. This assumption is supported by previous studies in similar populations and settings.

For example, Salman Saraç et al. (2024) calculated their sample size based on an effect size of 0.54 with similar statistical parameters in women with breast cancer. Mittal et al. (2025) reported a sample size calculation for anxiety and stress with 80% power and α = 0.05 in a population of undergraduate students. Moreover, Cantarero-Villanueva et al. (2016) used a comparable approach in cancer survivors, calculating sample size based on clinically meaningful differences with similar assumptions for power and significance level.

This evidence supports our choice of a medium effect size, ensuring adequate power without unnecessarily large samples. We have now included these references and this justification in the Methods section (see lines 97 to 103).

“Sample size was calculated using G*Power (Version 3.1.9.7), assuming a medium effect size (f = 0.25; Cohen’s d = 0.50), α = 0.05, and power = 0.80. This assumption is supported by studies on anxiety and related outcomes [19,20] and by similar sample size calculation approaches used in cancer survivor populations for other clinical variables [21]. A minimum of 72 participants was required, and 80 were ultimately recruited. Evaluations were conducted either at the Faculty of Health Sciences (University of Granada) or within the facilities of the collaborating associations”.

  • Salman Saraç F, Erkal İlhan S, Kutun S, Kutlutürkan S. The Effect of Informative Mobile App Use on Anxiety, Distress, and Quality of Life of Women With Breast Cancer. Eur J Breast Health. 2024 Jul 1;20(3):207-214. doi: 10.4274/ejbh.galenos.2024.2024-3-9. PMID: 39257013; PMCID: PMC11589289.

  • Mittal Nehal, Tyagi Niharika, Parvaiz Nida, Dhalla Nipun, Asthana Garima. Comparative Assessment of Stress and Anxiety among Medical and Nonmedical Undergraduate Students – A Cross-sectional Study. Indian Journal of Medical Specialities 16(1):p 60-64, Jan–Mar 2025. | DOI: 10.4103/injms.injms_197_24

  • Cantarero-Villanueva I, Sánchez-Jiménez A, Galiano-Castillo N, Díaz-Rodríguez L, Martín-Martín L, Arroyo-Morales M. Effectiveness of Lumbopelvic Exercise in Colon Cancer Survivors: A Randomized Controlled Clinical Trial. Med Sci Sports Exerc. 2016 Aug;48(8):1438-46. doi: 10.1249/MSS.0000000000000917. PMID: 27015381.
  1. Reviewer comment: “Statistical Analysis. Multiple Testing Correction: Given the large number of comparisons (many variables across groups), it would be appropriate to mention whether any correction for multiple testing was applied (e.g., Bonferroni, FDR). Otherwise, readers might question the risk of Type I error (false positives)”.

Author response:

We thank the reviewer for this insightful comment. We acknowledge the importance of addressing the risk of Type I error due to multiple comparisons. In response, we have now explicitly stated in the statistical analysis section that Bonferroni correction was applied where appropriate, particularly in post-hoc pairwise comparisons for normally distributed variables and in the analysis of non-parametric variables. This addition strengthens the robustness of our findings by ensuring a more conservative interpretation of statistical significance.

Additionally, to enhance clarity and improve the reader’s understanding of the statistical procedures used, we have reformulated the entire Statistical Analysis subsection using a bullet-point structure. This format aims to present the information in a more concise, transparent, and accessible way. The revised version can be found in the Methods section (see lines 230 to 252).

“2.3 Statistical analysis

Data analysis was performed using the Statistical Package for the Social Sciences (IBM SPSS Statistics for Windows, version 27.0, Armonk, NY, USA). The significance threshold was set at p < 0.05, with a 95% confidence interval (CI). The Kolmogorov-Smirnov test was applied to assess the normality of all variables, considering p > 0.05 as indicative of normal distribution.

  • Normally distributed variables (age, time since diagnosis, and time since first surgery) were analyzed using ANOVA to compare the two anxiety-level groups: lower (≤ 3.4) vs. higher (≥ 3.5). Results are reported as mean ± standard deviation. Post-hoc comparisons were adjusted using Bonferroni correction to account for multiple testing.

  • Non-normally distributed variables (mood state, non-categorized CRF, pain, self-perceived physical fitness, and HRQoL) were analyzed using the Mann-Whitney U test. Results are also reported as mean ± standard deviation. Bonferroni correction was applied where appropriate to minimize the risk of Type I error due to multiple comparisons.

  • Categorical variables, including other demographic, clinical, and medical characteristics, as well as categorized CRF and PA levels (based on predefined cut-off scores) [27,28,37–39], were analyzed using Chi-square tests and are presented as percentages”.

Results

  1. Reviewer comment: “Overloaded Tables: Tables 1–3 are too crowded and difficult to read. Suggested improvements: Split into sub-tables or focus only on significant variables in the main text. Move full demographic tables to Supplementary Material if allowed by journal guidelines”.

Author response:

We appreciate the reviewer’s comment regarding the readability of Tables 1–3. Upon reviewing the submitted version, we noticed that a formatting issue occurred during the manuscript submission process, which affected the row spacing of the tables and resulted in a visually overloaded appearance. We have corrected this issue in the revised version, restoring appropriate spacing and improving overall readability across all tables.

All tables now follow the formatting guidelines previously requested by the journal Life, as per our prior publications. To facilitate the editorial process, and based on our previous experience with the journal, we adopted this formatting approach from the outset.

Regarding the suggestion to split the tables or move some content to the supplementary material, we respectfully believe that this might not be the most effective strategy in this particular case. Tables 1–3 are structured according to their specific thematic focus: Table 1 includes sociodemographic and clinical data; Table 2 comprises the majority of outcome variables; and Table 3 exclusively presents HRQoL results from the EORTC QLQ-C30 and QLQ-BR23, which, due to the number of items, are traditionally presented separately in studies using these instruments. This structure reflects both content logic and consistency with prior literature.

Moreover, the majority of variables analyzed in Table 2 yielded statistically significant differences between anxiety groups. As such, separating only significant results into distinct tables would compromise the comprehensiveness and clarity of the comparative analysis, and potentially bias the perception of variable relevance.

Although physical activity was the only non-significant variable in Table 2, we opted to retain it within the same table to preserve conceptual coherence and avoid giving it diminished relevance. In fact, from this same table we extracted the cancer-related fatigue (CRF) variable—converted to percentage-based categories—to develop Figure 1, which highlights its distribution across anxiety groups. Creating a separate figure or placing the physical activity data in supplementary material would, in our opinion, imply a secondary role for this variable, which is not aligned with the comprehensive approach of our analysis.

Additionally, we aimed to maintain a balanced visual structure. The only two graphical elements included are: Figure 1, which displays the CRF variable in categorical format (percentage), and Figure 2, which shows the correlation matrix of key variables. Including additional visual elements for individual variables (e.g., physical activity) would risk overloading the manuscript and detracting from clarity.

Although the journal allows supplementary material, we believe that moving any of these key results to supplementary files could weaken their impact and misrepresent their relevance. These are not ancillary findings but essential components of the main analysis.

Finally, we recognize that the EORTC QLQ-C30 and BR23 instruments contribute to a more extended presentation of results due to the number of items. However, this is common in studies employing these tools, including previous publications by our group in Life and other peer-reviewed journals.

That said, we remain open to implementing further modifications should the reviewer or editorial team feel that additional adjustments are warranted. We hope the improved formatting and rationale provided will be sufficient to justify our current table and figure structure.

  1. Reviewer comment: “Figures: Figure 1 and Figure 2 captions need to be more descriptive (should stand alone without referring to text). Improve graphical quality (figures are a bit low-resolution in the file)”.

Author response:

We appreciate the reviewer’s feedback regarding the figures. While the resolution of the submitted images appears adequate on our end, we understand the importance of ensuring optimal clarity across platforms and devices. Therefore, we have enhanced the resolution and quality of both Figure 1 and Figure 2 to improve visual sharpness and overall readability.

The new images are provided in high-quality PNG format with the following specifications:

  • Resolution: 400 dpi
  • Dimensions: 5334 pixels (width) × 3000 pixels (height)

Regarding the suggestion to make figure captions more descriptive so they can “stand alone,” we understand this to mean that the captions should provide sufficient context and explanation for the figure to be interpreted independently of the main text. In response, we have revised both figure legends to include additional detail about the content, measures, and group distinctions presented. This ensures that the figures are more self-explanatory and accessible to readers reviewing them without immediate reference to the corresponding sections in the manuscript.

We hope these revisions address the reviewer’s concerns and contribute to a clearer and more informative presentation of the results (see lines 339 to 346 as well as 414 to 422).

“Figure 1. Distribution of cancer-related fatigue (CRF) severity by anxiety levels among long-term breast cancer survivors, expressed as percentages. Participants were stratified into low anxiety levels (VAS-A ≤ 3.4) and high anxiety levels (VAS-A ≥ 3.5). Fatigue severity was categorized using two cut-score systems: Cut-score A (No Fatigue: 0; Mild: 1–3; Moderate: 4–6; Severe: 7–10) and Cut-score B (No Fatigue: 0; Mild: 1–2; Moderate: 3–5; Severe: 6–10). Between-group comparisons were performed using the Mann–Whitney U test. Abbreviations: PFS: Piper Fatigue Scale, VAS-A: Visual Analogue Scale for Anxiety, n: sample size. Significant difference at p < 0.05; *p < 0.01”.

“Figure 2. Spearman’s correlation coefficients between anxiety levels and multiple health-related variables in long-term breast cancer survivors. Anxiety was assessed using the Visual Analogue Scale for Anxiety (VAS-A), analyzed as a continuous (non-categorized) variable. The figure displays the strength and direction of associations with measures of fatigue, pain, self-perceived fitness, and health-related quality of life. Abbreviations: VAS-A: Visual Analogue Scale for Anxiety, PFS: Piper Fatigue Scale, BPI: Brief Pain Inventory, IFIS: International Fitness Scale, QLQ-C30: EORTC Core Quality of Life Questionnaire, QLQ-BR23: Breast Cancer–Specific Module, SCC: Spearman’s Correlation Coefficient. *Significant correlation at p < 0.05; *p < 0.01”.

Discussion

  1. Reviewer comment: “Well-organized but verbose: Good integration of findings and literature. However, several paragraphs are too repetitive (e.g., discussion on CRF and anxiety). Suggested improvements: Summarize findings in a more concise way. Strengthen the "clinical implication" and "future research" sections”.

Author response:

We fully agree with this comment. We understand that, given the number of variables analyzed, some sections of the discussion were overly detailed and sometimes repetitive. In response, we have revised the paragraphs on CRF (lines 473 to 485) and pain (lines 487 to 498) to enhance clarity and conciseness, while preserving the integration of our findings with existing literature.

To strengthen the clinical relevance of the discussion, we have also added a brief clinical implication at the end of each paragraph addressing the key variables. We believe this approach improves the practical value of the findings and offers clearer guidance for future clinical applications.

“As for CRF, it is a common concern in LTBCSs and is closely linked to psychological distress, particularly anxiety and depression [3]. Consistent with previous studies, our findings show that participants high anxiety levels reported significantly greater CRF across all PFS domains, supporting a bi-directional relationship between fatigue and mood. Shared neurobiological mechanisms—such as HPA axis dysregulation and inflammation—may underlie this association [47]. Notably, the prevalence of moderate to severe CRF was substantially higher among LTBCSs with elevated anxiety, with 67.5% and 78.3% meeting severity thresholds under models A and B, respectively. These results reinforce evidence that anxiety exacerbates fatigue and its functional impact during survivorship [48]. As moderate CRF warrants clinical attention [30], screening for anxiety may be key to improving fatigue management. Future research should investigate targeted interventions, such as cognitive-behavioral therapy or structured exercise, to address anxiety-related CRF and enhance long-term quality of life”.

“Pain is a prevalent symptom among LTBCSs, with studies reporting post-treatment pain in approximately 46% of cases [49,50]. Our results show that those with high anxiety levels report significantly greater pain intensity in both the affected and unaffected arms (VAS), as well as higher pain severity and interference scores (BPI), compared to those with low anxiety. These findings align with evidence linking anxiety to increased pain perception in post-mastectomy women [3]. This relationship may be driven by shared neurobiological pathways, such as anxiety-related amplification of pain through dysregulated dopaminergic activity and overlapping pain-emotion neural circuits [51]. Moreover, the bidirectional interplay between pain and anxiety can create a self-reinforcing cycle of emotional and physical distress. Addressing anxiety may enhance pain management and improve HRQoL in LTBCSs. Integrated interventions targeting both symptoms should be prioritized in survivorship care”.

  1. Reviewer comment: “Causal Language Warning: Given the cross-sectional design, be careful not to imply causality ("Anxiety leads to worse outcomes"). Use language like "associated with" or "linked to".

Author response:

Thank you for your comment. We agree that causal language should be avoided in cross-sectional studies. Upon review, we identified only one instance in which causal wording ("lead to") was used in a statement derived from our own data. This has been revised to "may contribute to" to better reflect the associative nature of our findings (see lines 496 to 498).

“Addressing anxiety may enhance pain management and improve HRQoL in LTBCSs. Integrated interventions targeting both symptoms should be prioritized in survivorship care”.

The remaining uses of “lead to” appear in sentences supported by external references that explicitly reported causal relationships, and we have retained them accordingly (see lines 466 to 469 and 506 to 509, respectively). We appreciate your attention to this important distinction.

“Elevated anxiety may stem from concerns about cancer recurrence, leading to heightened emotional distress [45]. Additionally, the physical and social challenges encountered during survivorship, such as ongoing treatment side effects and changes in social roles, can exacerbate negative mood states [46]”.

“The interplay between anxiety and self-perception may be explained by cognitive distortions common in anxious individuals, leading to underestimation of their physical abilities despite a possible comparable objective performance [53]”.

  1. Reviewer comment: “Limitations. Good that limitations are acknowledged, but should also discuss: Lack of pre-cancer anxiety baseline. Limited generalizability to other ethnicities (only Spanish-speaking participants)”.

Author response:

Thank you for your valuable observation. We agree with your suggestion and have now incorporated both the absence of pre-cancer anxiety baseline and the limited generalizability due to the inclusion of only Spanish-speaking participants into the Limitations section of the manuscript. These points strengthen the discussion of the study’s constraints and will help guide future research directions (see lines 594 to 612).

“This study has several limitations that should be acknowledged. First, although the VAS-A provided a rapid and practical method for stratifying anxiety levels, the use of a single-item tool rather than a standardized multidimensional instrument (e.g., HADS, STAI, and the Generalized Anxiety Disorder-7 (GAD-7)) may limit the depth of the anxiety assessment. Nonetheless, the VAS-A has been shown to correlate with more comprehensive scales in diverse populations, and may serve as an efficient preliminary screening tool, particularly in clinical settings where time constraints preclude the use of lengthier instruments [24,25]. Moreover, we acknowledge that while our use of the VAS-A is grounded in previous work, it does not replace the depth of standardized multidimensional tools, and future studies should aim to validate these cut-offs further in specific cancer survivor populations. Second, the exclusive inclusion of female and Spanish-speaking LTBCSs limits the generalizability of findings to male survivors and non-Spanish-speaking or ethnically diverse groups. Third, the lack of information regarding pre-existing anxiety levels prior to cancer diagnosis restricts the ability to determine whether the observed anxiety is a result of the cancer experience or a pre-existing condition. Lastly, the cross-sectional nature of the study limits causal inferences, and although the cut-off points used were based on prior research in oncological populations [24], different thresholds might yield alternative results. Longitudinal studies are needed to clarify the directionality of these associations”.

  1. Reviewer comment: “References. Reference style is inconsistent (spacing errors, line breaks). Carefully reformat based on journal instructions”.

Author response:

We appreciate the reviewer’s observation regarding inconsistencies in the reference section.

Upon thorough revision, we identified that the presence of double spaces was due to a font-related issue in the original template (specifically, a subtype of Palatino used by default). After standardizing the font to the correct version of Palatino, these spacing inconsistencies were resolved.

Additionally, we have carefully revised and updated the reference list. In response to the reviewer’s suggestion to shorten the discussion, we have reduced the length of this section, which in turn led to the removal of some references that were no longer necessary. Furthermore, as suggested by another reviewer, we made a broader effort to reduce the overall number of references across the manuscript, which previously totaled 68.

We also:

  • Completed missing information in some entries (e.g., final page numbers);
  • Ensured internal consistency in formatting throughout the list.

Regarding the reference style, we maintained the format used in the original submission, which aligns with the journal’s editorial guidelines. According to the instructions:

“Your references may be in any style, provided that you use the consistent formatting throughout. It is essential to include author(s) name(s), journal or book title, article or chapter title (where required), year of publication, volume and issue (where appropriate) and pagination. DOI numbers (Digital Object Identifier) are not mandatory but highly encouraged.”

We have followed this guidance carefully and ensured that the references are now both complete and consistently formatted (see lines 663 to 807).

Minor concerns

  1. Reviewer comment: “Typographical Errors: Several sentences have double spaces between words. Some inconsistent use of periods after abbreviations (e.g., "CRF" vs "CRF."). Redundant phrases like "the level of anxiety level" — needs to be corrected to just "anxiety level".

Author response:

We appreciate the reviewer’s attention to detail.

  • We have carefully reviewed the manuscript and corrected all instances of double spaces between words.

  • Regarding the use of periods after abbreviations, we confirm that these only appear when the abbreviation coincides with the end of a sentence, not as part of the abbreviation itself. All abbreviations are now used consistently following scientific conventions (e.g., CRF, VAS-A).

  • Concerning the use of quotation marks, we have verified that their usage is intentional and appropriate. Quotation marks appear when referring to specific items or scales from standardized questionnaires. For example, when referring to domains such as "total CRF", "affective CRF", or "sensory CRF", we aim to highlight that these terms correspond directly to distinct components or items within validated instruments.

  • Finally, the redundant phrase "the level of anxiety level" has been corrected throughout the text. All such expressions have been reviewed and streamlined to avoid repetition and improve clarity.

We hope these corrections fully address the reviewer’s concerns.

  1. Reviewer comment: “Comments on the Quality of English Language. Language Polishing: Use more active voice instead of passive”:

 Example:

  • "Assessments were conducted" → "The research team conducted assessments".

Reword awkward sentences:

  • “Following the same protocol as hospital-referred participants, assessments were conducted at the association's premises” → "The same protocol used for hospital-referred participants was applied for assessments at association premises”.

Author response:

Thank you very much for your comment. We have carefully reviewed the entire manuscript to ensure that the use of passive voice does not hinder clarity. One of our co-authors, Maria Figueroa Mayordomo, is a native English speaker and has thoroughly revised the text to identify and correct any awkward or unclear sentences. We agree that some parts of the manuscript benefited from rewording, and we have made the necessary changes accordingly. Please find the revised sections detailed below.

We would also like to note that, in some instances, we intentionally used the passive voice to improve the flow of the text and reduce redundancy, particularly where repeating the subject in multiple consecutive sentences would have affected readability.

For women referred from the hospital, the oncology department was aware of the study's inclusion and exclusion criteria.” “The oncology department was aware of the study's inclusion and exclusion criteria for women referred from the hospital”. (See lines 105 to 106).

“Following the same protocol as hospital-referred participants, assessments were con-ducted at the association's premises.” The same protocol used for hospital-referred participants was applied for assessments at the association premises”. (See lines 116 to117).

“All assessments were consistently conducted by the same researcher, regardless of location.” The same researcher conducted all assessments, regardless of location”. (See lines 119 to 120).

“To classify participants into groups, anxiety levels were measured using an adapted version of Visual Analogue Scale (VAS) for Anxiety (VAS-A).” → “Anxiety levels were measured to classify participants into groups using an adapted version of the Visual Analogue Scale (VAS) for Anxiety (VAS-A)”. (See lines 129 to 130).

“This tool has shown to be reliable for assessing anxiety in BC patients [26].” → “Previous studies have shown this tool to be reliable for assessing anxiety in BC patients [26].” (See line 152).

“Data collection was carried out through structured interviews using a tailored questionnaire aimed at gathering sociodemographic and clinical information.”  → “Structured interviews were conducted using a tailored questionnaire to gather sociodemographic and clinical information” (See lines 158 to 159).

“CRF was assessed using the Piper Fatigue Scale (PFS), a 22-item instrument that evaluates four key dimensions:…”“The Piper Fatigue Scale (PFS), a 22-item instrument, was used to assess CRF across four key dimensions:…” (See lines 174 to 175).

“Energy expenditure was calculated by multiplying the reported weekly duration (in hours) of each activity by its corresponding Metabolic Equivalent of Task (MET) value [38]”“To calculate energy expenditure, the reported weekly duration (in hours) of each activity was multiplied by its corresponding Metabolic Equivalent of Task (MET) value [38].” (See lines 211 to 213).

The author responsible for correspondence is:

Clara Pujol Fuentes
FIBIO Research Group,

Department of Physiotherapy,
Faculty of Health Sciences, European University of Valencia,
46010, Valencia, Spain

+34 661075990

clara.pujol@universidadeuropea.es

Sincerely,

Clara Pujol Fuentes

Reviewer 2 Report

Comments and Suggestions for Authors

I am pleased to review your valuable manuscript. Your study examines anxiety and health status among long-term breast cancer survivors, which is a topic of great interest to many clinicians who work with breast cancer patients. I have a few comments on your article that I hope will contribute to improving your study. Thank you.

  1. You included only Spanish participants in your study. Therefore, your title should include "in Spain." Please reconsider this and make necessary adjustments.

  1. In the introduction section, you thoroughly addressed the significance of your study. I have no comments on this section.

  1. I noticed that the font style in the methods section of the manuscript has changed. These comments do not relate to the content of your study, but it's essential to maintain consistency in font throughout the document. Please review and revise as necessary to ensure coherence in font characteristics.

  1. In the results section, not all tables present the scores of the full 80 participants. As a reader, I would like to understand the overall tendencies among the participants before comparing the groups. Please reconsider this and revise it as needed.

  1. In the discussion section, you should include comparative opinions that contrast the overall participants' tendency with the tendencies of the comparing groups. Because your definition of higher anxiety and lower anxiety was only 1 mm different among the groups, please reconsider and revise here, depending on your necessity.

  1. You have 68 references in your study, which seems excessive. Please reconsider and revise the reference list better to reflect the significance of the references in your work.

Author Response

Clara Pujol Fuentes

E-mail address

clara.pujol@universidadeuropea.es

+34 661075990
Full postal address

FIBIO Research Group,

Department of Physiotherapy,
Faculty of Health Sciences, European University of Valencia,
46010, Valencia, Spain

Editorial Reviewer 2

Life

2 June 2025

Dear Reviewer 2,

Please find below the answers to each of your contributions

“I am pleased to review your valuable manuscript. Your study examines anxiety and health status among long-term breast cancer survivors, which is a topic of great interest to many clinicians who work with breast cancer patients. I have a few comments on your article that I hope will contribute to improving your study. Thank you”.

Author response:

First of all, we would like to thank you for your words and the time dedicated to the understanding and improvement of this scientific work. In this way, and from here on, all the answers are detailed, individually and by sections, to each of your suggestions or comments.

With this in mind, we believe that, thanks to the reviewer's contributions and suggestions, together with all our responses, this scientific research is now much easier to read and understand.

General comments

  1. Reviewer comment: “You included only Spanish participants in your study. Therefore, your title should include "in Spain." Please reconsider this and make necessary adjustments”.

Author response:

Thank you for this valuable suggestion. We have revised the title to reflect the geographic location of the study participants as follows:

“Influence and Predictors of Anxiety on Health Status ≥ 5 Years Beyond Breast Cancer Diagnosis in Spain: A Cross-Sectional Study.”

This change enhances the clarity and contextual framing of the study population.

  1. Reviewer comment: “In the introduction section, you thoroughly addressed the significance of your study. I have no comments on this section”.

Author response:

We sincerely thank the reviewer for the positive feedback regarding the Introduction section and for recognizing the relevance and clarity of the background provided.

However, in response to Reviewer 1's suggestion to reduce redundancy—particularly regarding the differentiation between anxiety and depression—we have made several modifications to streamline the narrative. Specifically, we shortened certain background paragraphs and emphasized the knowledge gap earlier and more concisely, as recommended. These adjustments aimed to maintain the scientific rigor while improving the overall clarity and focus of the section.

  1. Reviewer comment: “I noticed that the font style in the methods section of the manuscript has changed. These comments do not relate to the content of your study, but it's essential to maintain consistency in font throughout the document. Please review and revise as necessary to ensure coherence in font characteristics”.

Author response:

We appreciate the reviewer’s observation regarding the inconsistency in font style. Reviewer 1 also raised a similar point, and based on the feedback from both reviewers, we have thoroughly revised the entire manuscript to ensure uniformity in font characteristics.

It is likely that, due to the collaborative nature of the manuscript preparation and the presence of two similar font types (Palatino and Palatino Linotype), some inadvertent inconsistencies were introduced. We sincerely apologize for any inconvenience this may have caused and have now corrected these issues to maintain coherence and adhere to editorial standards.

  1. Reviewer comment: “In the results section, not all tables present the scores of the full 80 participants. As a reader, I would like to understand the overall tendencies among the participants before comparing the groups. Please reconsider this and revise it as needed”.

Author response:

Thank you very much for your comment. We have thoroughly reviewed all the tables and can confirm that data from the full sample of 80 participants are included across all of them. The total sample is consistently divided into two subgroups: 43 participants with lower anxiety levels and 37 participants with higher anxiety levels. All items from all questionnaires were fully completed by the 80 participants, and no missing data were found.

The only value that is not shown is the Cohen’s d for the MLTPA variable in Table 2. This is because MLTPA is expressed as a percentage, and Cohen’s d is not typically calculated for categorical or percentage-based variables, in line with standard statistical practice.

If there is a specific part of the results the reviewer believes may still be unclear or missing, we would sincerely appreciate further clarification so that we can address it precisely. However, after carefully checking all tables, we confirm that no data are missing from the presented results.

  1. Reviewer comment: “In the discussion section, you should include comparative opinions that contrast the overall participants' tendency with the tendencies of the comparing groups. Because your definition of higher anxiety and lower anxiety was only 1 mm different among the groups, please reconsider and revise here, depending on your necessity”.

Author response:

Thank you for your insightful comment. We have addressed your suggestion by adding the following paragraph at the beginning of the Discussion section (see lines 449 to 458):

"Although the cut-off point used to define higher and lower anxiety (3.5 vs. 3.4) may appear narrow, it was derived from a previous validation study using the VAS-A in oncological settings [24], where scores ≥3.5 have been shown to reflect clinically relevant anxiety. In our cohort, this threshold revealed consistent and meaningful differences in multiple health domains. Therefore, despite the subtle numerical distinction, the stratification proved useful for identifying long-term breast cancer survivors (LTBCSs) at greater risk for symptom burden and poorer health outcomes. However, this cut-off should not be interpreted as a definitive diagnostic threshold. Rather, it may serve as an initial red flag in time-constrained clinical settings, prompting more comprehensive psychological assessments or referral to mental health professionals when anxiety scores exceed 3.5."

We believe this clarification better contextualizes the cut-off point and its clinical relevance, while acknowledging its limitations.

  1. Reviewer comment: “You have 68 references in your study, which seems excessive. Please reconsider and revise the reference list better to reflect the significance of the references in your work”.

Author response:

We appreciate the reviewer’s observation regarding the number of references included in our manuscript. We acknowledge that 68 references may seem extensive; however, this is partly due to the use of multiple validated instruments to assess different variables in our study. Each instrument requires citation of its original validation studies, cut-off points, reliability, and other relevant psychometric properties to ensure scientific rigor and transparency.

That said, we have made a concerted effort to reduce the number of references, especially in the Introduction and Discussion sections. Initially, many statements were supported by 3 to 4 references to provide the strongest possible evidence. We have now carefully screened and selected the most important and recent references to better reflect the current state of knowledge. With a final reference count of "59", we hope the manuscript now appears more concise and focused, while maintaining appropriate scientific support.

On the other hand, and following the request of the other reviewer, those 59 references already include four additional sources that were necessary to support some of the proposed recommendations. These new references correspond to numbers 19, 20, 21, and 25 in the document.

We remain open to further suggestions on this matter.

The author responsible for correspondence is:

Clara Pujol Fuentes
FIBIO Research Group,

Department of Physiotherapy,
Faculty of Health Sciences, European University of Valencia,
46010, Valencia, Spain

+34 661075990

clara.pujol@universidadeuropea.es

Sincerely,

Clara Pujol Fuentes

Round 2

Reviewer 1 Report

Comments and Suggestions for Authors

Upon review, it appears that the authors have already addressed the suggested revisions.

Author Response

Clara Pujol Fuentes

E-mail address

clara.pujol@universidadeuropea.es

+34 661075990
Full postal address

FIBIO Research Group,

Department of Physiotherapy,
Faculty of Health Sciences, European University of Valencia,
46010, Valencia, Spain

Reviewer 2

Life

4 June 2025

Dear Reviewer 2,

Please find below the answers to each of your contributions

“Upon review, it appears that the authors have already addressed the suggested revisions”.

Author response:

We sincerely appreciate the time and effort you have dedicated to reviewing the revisions submitted following the initial round of comments. However, we would like to kindly request clarification regarding the designation of the manuscript as “can be improved.” Should you have any additional suggestions or comments, we remain fully open to receiving your valuable feedback.

Sincerely,

Clara Pujol Fuentes
